



**Socio-meteorology: flood prediction, social preparedness, and cry wolf effects**
Yohei Sawada[1], Rin Kanai[2], and Hitomu Kotani[1,3]
[1] Institute of Engineering Innovation, the University of Tokyo, Tokyo, Japan
[2] Department of Civil Engineering, the University of Tokyo, Tokyo, Japan
[3] Department of Urban Management, Kyoto University, Kyoto, Japan
Corresponding author: Y. Sawada, Institute of Engineering Innovation, the University of
Tokyo, Tokyo, Japan, 2-11-6, Yayoi, Bunkyo-ku, Tokyo, Japan, yohei.sawada@sogo.t.u-
tokyo.ac.jp





**Abstract**
To improve the efficiency of flood early warning systems (FEWS), it is important to
understand the interactions between natural and social systems. The high level of trust in
authorities and experts is necessary to improve the likeliness of individuals to take
preparedness actions responding to warnings. Despite a lot of efforts to develop the
dynamic model of human and water in socio-hydrology, no socio-hydrological models
explicitly simulate social collective trust in FEWS. Here we develop the stylized model
to simulate the interactions of flood, social collective memory, social collective trust in
FEWS, and preparedness actions responding to warnings by extending the existing socio-
hydrological model. We realistically simulate the cry wolf effect, in which many false
alarms undermine the credibility of the early warning systems and make it difficult to
induce preparedness actions. We found (1) considering the dynamics of social collective
trust in FEWS is more important in the technological society with infrequent flood events
than in the green society with frequent flood events; (2) as the natural scientific skill to
predict flood events is improved, the efficiency of FEWS gets more sensitive to the
behavior of social collective trust, so that forecasters need to determine their warning
threshold by considering the social aspects.





## 1. Introduction

The number of severe flood events is expected to increase in many regions due to climate

change (Hirabayashi et al. 2013, 2021). Based on the advances of weather forecasting

(e.g., Bauer et al. 2015; Miyoshi et al. 2016; Sawada et al. 2019) and hydrodynamic

modeling (e.g., Yamazaki et al. 2011; Trigg et al. 2016), Flood Early Warning Systems

(FEWS) have become the promising tool to efficiently mitigate the damage of severe

floods. However, to maximize the potential of FEWS, it is crucially important to

understand the interactions between flood and social systems. The likeliness of

individuals to take preparedness actions responding to flood warnings strongly depends

on the individual's risk perception which is controlled by the complex interaction between

natural hazards and stakeholders (Wachinger et al. 2013).

In the literature of weather forecasting, the "cry wolf effect" has been intensively

investigated as an important interaction between weather prediction and social systems.

In Aesop's fable, the "The Boy who Cried Wolf", a young boy repeatedly tricks

neighboring villagers into believing that a wolf is attacking the sheep. When a wolf

actually appears and the young boy seriously calls for help, the villagers no longer trust



the warning and fail to protect their sheep. Many false alarms undermine the credibility
of the early warning systems. The cry wolf effect on mitigation and protection actions
against meteorological disasters has been investigated in economics, sociology, and
psychology. Simmons and Sutter (2009) performed econometric analysis of a disaster
database and revealed that tornadoes that occurred in areas with higher false alarm ratio
killed and injured more people. Ripberger et al. (2015) performed a web-based
questionnaire survey and revealed that subjective perceptions of warning system's
accuracy are systematically related to trust in a weather agency and stated responses to
warnings. Trainor et al. (2015) performed large-scale telephone interviews and revealed
the significant relationship between actual false alarm ratio and behavioral responses to
tornado warnings. They also found that there is a wide variation in public definition of
false alarms and actual false alarm ratio does not predict perception of false alarm ratio,
which illustrated the significant complexity associated with the analysis of false alarms.
Although Trainor et al. (2015) could not find the significant relationship between
perceived false alarm ratio and responses to warnings, Jauernic and van den Broeke
(2017) revealed that the odds of students initialing sheltering decreases nearly 1% for
every 1% increase in perceived false alarm ratio based on their online questionnaire
survey of 640 undergraduate students. While these previous works supported the cry wolf





effect as an important factor to be considered for the design of warning systems, many
existing studies discussed the myth of cry wolf effects implying that they do not exist.
For example, LeClerc and Joslyn (2015) performed a psychological experiment in which
participants decided whether to apply salt brine to a town's roads to prevent icing
according to weather forecasting. In their experiment, the effects of false alarms are so
small that they found no evidence suggesting lowering false alarm ratio significantly
increases compliance with weather warnings. Lim et al. (2019) performed an online
questionnaire survey and found no significant relationship between actual false alarm
ratio and responses to warnings. In addition, they found that the increase of perceived
false alarm ratio enhanced protective behavior, which contradicted the other works.
Although the existence of the cry wolf effect is still debatable, the warning threshold of
the actual weather warning systems can be justified only if the cry wolf effect is
considered (Roulston and Smith 2003). It is crucially important to understand the effect
of false alarms on behavioral responses to warnings to design efficient weather warning
systems.

Socio-hydrology is an emerging research field to contribute to understanding the
interactions between flood and social systems (Sivapalan et al. 2012, 2014; Di



Baldassarre et al. 2019). The primary approach of socio-hydrology is to develop the
dynamic model of water and human. Many socio-hydrological models used social
preparedness as a key driver of human-water interactions (e.g., Di Baldassarre et al. 2013;
Viglione et al. 2014; Ciullo et al. 2017; Yu et al. 2017; Albertini et al. 2020). The
pioneering work of Girons Lopez et al. (2017) revealed the effect of social preparedness
on the efficiency of FEWS. Their main finding is that social preparedness is an important
factor for flood loss mitigation especially when the accuracy of the forecasting system is
limited. However, to our best knowledge, the existing socio-hydrological models
simulated social preparedness as a function of social collective memory or personal
experience of past disasters, and they considered no effect of trust in authorities and
experts. Therefore, the cry wolf effect cannot be analyzed in the existing models. The
systematic review of Wachinger et al (2013) indicated that both personal experience of
past disasters and trust in authorities and experts have the substantial impact on risk
perception. It is crucially important to include the social collective trust in FEWS in the
socio-hydrological model to improve the design of FEWS considering social system
dynamics.



The aim of this study is to develop the stylized model of the responses of social systems
to FEWS as the simple extension of Girons Lopez et al. (2017). By modeling the
dynamics of social collective trust in FEWS as a function of the recent success and failure
of the forecasting system, we realistically simulate the cry wolf effect. By analyzing our
newly developed model, we provide useful implication to maximize the potential of
FEWS considering social system dynamics.

**2. Model**
Here we slightly modified the model proposed by Girons Lopez et al. (2017). For brevity,
the detailed explanation of equations shared with Girons Lopez et al. (2017) is omitted in
this paper. See Gironz Lopez et al. (2017) and references therein for the complete
description.

A synthetic time series of river discharge is generated. Following Girons Lopez et al.
(2017), a simple bivariate gamma distribution, $\Gamma$, is used:
$Q \sim \Gamma(\kappa_c, \theta_c)$   (1)
where Q is maximum annual flow. The bivariate gamma distribution is characterized by
shape $\kappa_c$ and scale $\theta_c$.






This maximum annual flow, Q, is forecasted. In our model, the ensemble flood forecasting
system (e.g., Cloke and Hornberger 2009) is installed and the probabilistic forecast can
be issued. The forecast probability distribution, $F$, is calculated by the following:
$F \sim N(Q + N(\mu_m, \sigma_m^2), N(\mu_v, \sigma_v^2))$  (2)
where $N(.)$ is the Gaussian distribution, $N(\mu_m, \sigma_m^2)$ controls the prediction accuracy,
and $N(\mu_v, \sigma_v^2)$ controls the prediction precision. While Girons Lopez et al. (2017)
changes $\mu_m$ in their simulation, we set $\mu_m = 0$ assuming the forecast is unbiased.
While Girons Lopez et al. (2017) used the bivariate gamma distribution to model the
prediction precision, we used the Gaussian distribution to make it easier to interpret
results.

There is a damage threshold, $\delta$, which is the proxy of levee height. When $Q > \delta$, flood
occurs. The forecast system calculates the probability of river discharge exceeding $\delta$
and issues a warning if this probability of exceedance, $P$, is larger than a predefined
probability threshold, $\pi$. Table 1 summaries four different outcomes of forecasting: true
positive, false positive, false negative, and true negative. When forecasters choose lower
$\pi$, they issue many warnings with low forecasted probability of flooding, which inevitably



increases false alarms. When forecasters choose higher $\pi$, they can reduce the number of
false alarms by issuing the smaller number of warnings, which inevitably increases
missed events.

Based on these four different outcomes shown in Table 1, damages and costs are
calculated. Flood damage is assumed to be negligible when river discharge is smaller than
a damage threshold (i.e. $Q < \delta$). When $Q \geq \delta$, the damage function is defined as a
simple exponential function, which is often used in the socio-hydrological literature (e.g.,
Di Baldassarre et al. 2013):
$$D_Q = \begin{cases} 0 & (Q < \delta) \\ 1 - e^{-\frac{Q-\delta}{\beta}} & (Q \geq \delta) \end{cases} \quad (3)$$
where $D_Q$ is damage, $\beta$ is a model parameter. If a flood event is successfully forecasted
and a warning is issued (i.e. $P \geq \pi$), this damage is mitigated by preparedness actions.
How much damage can be mitigated depends on social preparedness, $P_r$. The mitigated
damage (called residual damage in Girons Lopez et al. (2017)), $D_r$, is calculated by the
following:
$$D_r = D_Q e^{-P_r \ln(\frac{1}{\alpha_0})} \quad (4)$$
where $\alpha_0$ is a model parameter which determines the minimum possible damage. In
summary, the flood damage, $D$, can be described by equation (5):


$$D = \begin{cases} 0 & (Q < \delta) \\ 1 - e^{-\frac{Q-\delta}{\beta}} & (Q \geq \delta \ and \ P < \pi) \\ \left(1 - e^{-\frac{Q-\delta}{\beta}}\right) e^{-P_r \ln\left(\frac{1}{\alpha_0}\right)} & (Q \geq \delta \ and \ P \geq \pi) \end{cases} \quad (5)$$

Whenever a warning is issued, the cost, $C$, arises from mitigation and protection actions.
Following Girons Lopez et al. (2017), we assumed that the cost is calculated by:
$$C = \begin{cases} 0 & P < \pi \\ \eta Q & P \geq \pi \end{cases} \quad (6)$$
where $\eta$ is a parameter.

The dynamics of social preparedness, $P_r$, in this study is different from Girons Lopez et
al. (2017). We assumed that the social preparedness consists of social collective memory
and social collective trust in FEWS:
$P_r(t) = \gamma E(t) + (1 - \gamma)T(t)$ (7)
where $E(t)$ and $T(t)$ are social collective memory and social collective trust in FEWS
at time $t$, respectively. $\gamma$ is a model parameter that weights $E(t)$ and $T(t)$. In many
socio-hydrological models, social collective memory is driven by the recency of past
flood experience. Following Girons Lopez et al. (2017), the dynamics of social collective
memory is described by the following:
$$E(t + 1) = \begin{cases} E(t) - \lambda E(t) & (D = 0) \\ E(t) + \chi D & (D > 0) \end{cases} \quad (8)$$



where $\lambda$ and $\chi$ are model parameters.

We assumed that social collective trust in FEWS is affected by the recent accuracy of
FEWS. Previous studies pointed out that the recent forecast accuracy and false alarm ratio
affected the performance of preparedness actions (Simmons and Sutter 2009; Trainor et
al. 2015; Ripberger et al. 2015; Jauernic and van den Broeke 2017). It is reasonable to
assume that trust in FEWS increases (decreases) when prediction succeeds (fails)
(Wachinger et al. 2013). We propose the following simple equation to describe the
dynamics of social collective trust in FEWS:
$$T(t+1) = \begin{cases} T(t) & for\ true\ negative \\ T(t) + \tau_{TP} & for\ true\ positive \\ T(t) - \tau_{FN} & for\ false\ negative \\ T(t) - \tau_{FP} & for\ false\ positive \end{cases} \quad (9)$$
where $\tau_{TP}$, $\tau_{FN}$, and $\tau_{FP}$, are positive parameters. By changing the value of these
parameters, we can change the sensitivity of social collective trust in FEWS to the
accuracy of FEWS. We will analyze the behavior of our model associated with several
different combinations of these three parameters.

In our equations (7-9), we can consider both social collective memory and social
collective trust to analyze behavioral responses to warnings. For instance, please assume





that a severe flood occurs and substantially damages a community, and this flood events
cannot be predicted. In this case, social collective memory increases due to the large
damage (equation (8)). This increase of social collective memory $E(t)$ contributes to
increasing social preparedness towards the next severe flood events (equation (7)).
However, the failure of predicting this flood events decreases social collective trust in
FEWS and authorities related to warning systems (equation (9)), which negatively
impacts to the capability of a community to deal with the next flood events by decreasing
social preparedness (equation (7)).

If social preparedness is determined only by social collective memory as Girons Lopez et
al (2017) proposed, social preparedness constantly decreases and goes to 0 when no
floods occur for a long while. In our proposed model, high social collective trust in FEWS
can maintain the high level of social preparedness even if a community completely loses
past flood experiences (equation (7)). However, if a weather agency repeatedly issues
false alarms, social collective trust in FEWS decreases (equation (9)), which negatively
impacts to social preparedness (equation (7)). Therefore, the dynamics of social
preparedness in our proposed model is greatly different from Girons Lopez et al. (2017).



Many of the model parameters are fixed in our analysis. Table 2 summarizes the
description and values of the fixed parameters. Some parameters are changed in our
analysis to check their sensitivity to the performance of FEWS. Those parameters are
explained in the next section.

**3. Experiment design**
**3.1. Metrices**
We used several metrics to evaluate the performance of FEWS. The purpose of FEWS
is to reduce the total loss $(D + C)$. We used the relative loss as Girons Lopez et al. (2017)
did. The relative loss, $L_r$, is defined by equation (10):
$L_r = \frac{L_{FEWS}}{L_{noFEWS}}$    (10)
We performed the long-term (1000-year) numerical simulation by solving equations (1-
9) and calculated the total loss, $L_{FEWS}$. We also performed the simulation without FEWS,
in which flood damage is always calculated by equation (3) and $D$ is always equal to $D_Q$.
The total loss of this additional simulation is defined as $L_{noFEWS}$. The relative loss
measures the efficiency of FEWS.



In addition to relative loss, we used hit rate, false alarm ratio, and threat score to evaluate
the prediction accuracy, which is not related to social system dynamics. They are defined
by equations (11-13):
$hit\ rate = \frac{O_{TP}}{O_{TP}+O_{FN}}$    (11)
$false\ alarm\ ratio = \frac{O_{FP}}{O_{FP}+O_{TP}}$ (12)
$threat\ score = \frac{O_{TP}}{O_{TP}+O_{FP}+O_{FN}}$    (13)
where $O_{TP}$, $O_{FN}$, and $O_{FP}$ are the total number of true positive, false negative, and false
positive events, respectively.


**3.2. Simulation Settings**
We firstly compared the original model proposed by Girons Lopez et al. (2017) with our
modified model. When we set $\gamma = 1$ in equation (7), our model reduces to Girons Lopez
et al. (2017) since we have no contributions of social collective trust in FEWS to social
preparedness. In this paper, this original model is hereafter called the GL model. On the
other hand, when we set $\gamma = 0.5$ in equation (7), our model considers both social
collective memory and social collective trust in FEWS with same weights to calculate
social preparedness. This new model is hereafter called the SKK model.

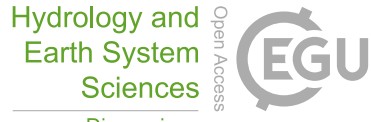


In the experiment 1, the timeseries of state variables of the two models are compared to
demonstrate how differently the SKK and GL models work. The parameter variables in
the experiment 1 are shown in Table 3.

We mainly focused on the relationship between relative loss and a predefined probability
threshold, $\pi$. This warning threshold is important for forecasters to determine whether
they require general citizens to take preparedness actions. In the experiment 2, we used
the same damage threshold, $\delta$, as Girons Lopez et al (2017) and compared the
relationship between relative loss and predefined warning thresholds in the GL model
with that in the SKK model under the different prediction skills and the cost parameter $\eta$.
The settings of the parameters in the experiment 2 can be found in Table 4.

In the experiment 3, we also compared the GL and SKK models under different damage
thresholds, $\delta$. In socio-hydrology, previous works focused on the difference between
"green" and "technological" society (Ciullo et al. 2017). In green society, the flood
protection level is so low that many flood events occur, which increases social collective
memory of flood events. In technological society, the flood protection level is high. Since





flood events occur less frequently in the technological society, the high level of social
collective memory cannot be maintained. By changing the damage threshold, we
analyzed how differently the GL and SKK models behave in the different society. The
settings of the parameters in the experiment 3 can be found in Table 5.

In the experiment 4, we analyzed only the SKK model. The primary purpose of this
experiment 4 is to find the optimal predefined warning threshold, which minimizes
relative loss, in not only different society and prediction accuracy but also different
combinations of parameters related to the dynamics of social collective trust in FEWS
(i.e., $\tau_{TP}, \tau_{FN}$, and, $\tau_{FP}$ in equation (9)). The settings of the parameters in the
experiment 4 can be found in Table 6.

In experiments 2–4, we performed the 250-member Monte-Carlo simulation by randomly
perturbing a predefined probability threshold, $\pi$, and the initial conditions of social
collective memory and social collective trust in FEWS. We analyzed the sensitivity of the
efficiency of FEWS to predefined warning thresholds.



## 4. Results


Figure 1 shows the time series of social preparedness of the GL and SKK models in the
experiment 1 (see Table 3). In the GL model (Figure 1a), social preparedness (black line)
increases when flood occurs (red and green bars) and is not affected by false alarms (blue
bars). In the SKK model (Figure 1b), false alarms negatively impact social preparedness
by reducing social collective trust in FEWS (pink line). From $t = 430$ to $t = 440$,
consecutive false alarms substantially decrease social collective trust in FEWS and social
preparedness, so that the damage of severe flood at $t = 452$ in the SKK model is larger
than that in the GL model despite the accurate warning being issued. It is the cry wolf
effect.

Figure 2a shows the relationship between relative loss and predefined warning thresholds
simulated by the GL model in the experiment 2 (see Table 4). We firstly assumed that
there is no cost of the mitigation and protection action and is the relatively accurate
prediction system (the experiment 2.1; see Table 4). In this case, FEWS can minimize the
relative loss with the extremely small predefined warning thresholds (blue line). When
we degrade the prediction skill (the experiment 2.2; see Table 4), forecasters still maintain
the same level of relative loss by setting low (or zero) predefined warning thresholds



issuing many false alarms (orange line). It is apparently unrealistic. In the framework of
the GL model, this unrealistic model's behavior can be eliminated by setting the high cost
of the mitigation and protection action responding to the issued warning. When we
assume the high cost of preparedness actions (the experiment 2.3; see Table 4), the small
predefined warning threshold induces high relative loss (green line). Forecasters need to
avoid issuing false alarms when the cost which should be paid with false alarms is large.

The SKK model can give different explanation of the avoidance of false alarms. Figure
2b shows the relationship between relative loss and predefined warning thresholds
simulated by the SKK model in the experiment 2 (see Table 4). Although we assumed no
cost and an accurate prediction system (the experiment 2.4; see Table 4), forecasters need
to avoid issuing false alarms by the relatively high predefined warning thresholds to
minimize relative loss (blue line). Due to the cry wolf effect found in Figure 1b,
forecasters need to decrease the number of false alarms to mitigate the damage of flooding
even if there were no cost of false alarms. In other words, forecasters in the SKK model
need to pay "implicit cost" of false alarms because false alarms induce not only the cost
of mitigation and protection actions for nothing at the current time but also the increase
of damages of the future floods by reducing the social collective trust and preparedness.



When we degrade the prediction accuracy (the experiment 2.5; see Table 4), relative loss
is more sensitive to predefined warning thresholds (orange line) because the selection of
the threshold is more important to accurately detect flood events and reduce the number
of false alarms when the prediction is more inaccurate and uncertain. When we consider
the high cost of mitigation and protection actions (the experiment 2.6; see Table 4), small
predefined warning thresholds further increase relative loss (green line).

Figure 3a compares the GL and SKK models in the green society. In the previous
experiments 1 and 2, the damage threshold, $\delta$, is set to 0.35, which is same as Girons
Lopez et al. (2017). In the experiments 3.1 and 3.2 (see Table 5), the damage threshold is
reduced to 0.20, so that the number of flood events increases. In this case, the GL and
SKK models behave similarly. Figure 3c shows time-averaged social collective memory,
social collective trust in FEWS, and social preparedness as functions of predefined
warning thresholds. In the green society, frequent flood events make social collective
memory high. In addition, it is easy to maintain the high social collective trust in FEWS
since there are many opportunities to gain trust when flood frequently occurs. Therefore,
both social collective memory and social collective trust in FEWS are large in the green



society. Although the GL model neglect the social collective trust in FEWS to calculate
social preparedness, the social preparedness of both GL and SKK models is high

On the other hand, the GL and SKK models work more differently in the technological
society than the green society. The damage threshold, $\delta$, is increased to 0.45 in the
experiments 3.3 and 3.4 (see Table 5), so that the number of flood events is smaller than
Girons Lopez et al. (2017). Figure 3b indicates that the relationship between relative loss
and predefined warning thresholds in the GL model is substantially different from that in
the SKK model. The SKK model produces smaller relative loss than the GL model when
the appropriate predefined warning threshold is chosen. The sensitivity of relative loss to
predefined warning thresholds is larger in the technological society than the green society.
Figure 3d indicates that it is difficult to maintain the high level of social collective
memory in the technological society, so that considering social collective trust in FEWS
can increase social preparedness. In addition, the choice of a predefined warning
threshold is more important to maintain the high level of social collective trust in the
technological society than the green society.

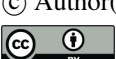



In the experiment 4, we further analyze the SKK model to discuss the optimal predefined
warning threshold and to provide the useful implication for the design of FEWS in the
various kind of social systems. We have three sets of parameters in equation (9) (see also
Table 6). The first set of parameters is same as the experiments 1-3. Changes in social
collective trust by false negative and false positive are same ($\tau_{FN} = \tau_{FP}$). In the second
set of parameters, we assume social collective trust substantially decreases by false
positive (false alarms) ($\tau_{FN} < \tau_{FP}$): $[\tau_{TP}, \tau_{FN}, \tau_{FP}] = [0.1, 0.1, 0.8]$. In the third set of
parameters, we assume social collective trust substantially decreases when forecasters
miss a flood event ($\tau_{FN} > \tau_{FP}$): $[\tau_{TP}, \tau_{FN}, \tau_{FP}] = [0.1, 0.8, 0.1]$. The blue, orange, and
green lines in Figures 4a-4d show that the optimal predefined warning threshold depends
on how social collective trust is affected by false alarms and missed events. When social
collective trust is affected by false alarms more substantially than missed events (orange
lines), forecasters need to have relatively high predefined warning thresholds to maintain
the high level of social collective trust (see Figures 4e-h) and minimize relative loss.
Figures 4a-4d also shows that the differences of optimal predefined warning thresholds
in three sets of parameters become larger as forecasts become accurate. The optimal
predefined thresholds are bounded by the range in which the high threat scores can be
obtained (see Figures 4i-4l).   Thus, more accurate prediction systems make it more





important to change the predefined warning threshold according to the dynamics of social
collective trust. It implies that forecasters need to prioritize the meteorologically accurate
forecasting by maximizing threat scores. Then, they have a room for improvement to
change their warning thresholds based on the dynamics of social collective trust in FEWS.

**5. Discussion and conclusions**
In this study, we included the dynamics of social collective trust in FEWS into the existing
socio-hydrological model. By formulating social preparedness as a function of social
collective trust as well as social collective memory, we realistically simulate the cry wolf
effect, in which many false alarms undermine the credibility of the early warning systems.
Please note that the previous version of the model proposed by Girons Lopez et al. (2017)
cannot do it. Although our model is simple and stylized, we can provide useful implication
to improve the design of FEWS. First, considering the dynamics of social collective trust
in FEWS is more important in the technological society with infrequent flood events than
in the green society with frequent flood events. Second, as the natural scientific skill to
predict flood is improved, the efficiency of FEWS gets more sensitive to the behavior of
social collective trust, so that forecasters need to determine their forecasting threshold by
considering the social aspects.




Although our model is the small extension of Girons Lopez et al. (2017), the implication
of our study is completely different from Girons Lopez et al. (2017). Girons Lopez et al.
(2017) mainly focused on the influence of the recency of flood experience on social
preparedness and the efficiency of FEWS. Since their social preparedness is determined
only by the flood experiences and they did not consider social collective trust in FEWS
and weather agencies, the outcome of prediction did not directly influence the people's
behavior in the model of Girons Lopez et al. (2017). By formulating social preparedness
as a function of both social collective memory and trust, we could evaluate the effects of
missed events and false alarms on preparedness actions. We contributed to connecting the
modeling approaches of system dynamics in socio-hydrology to the existing literature
about complex human behaviors against disaster warnings such as cry wolf effects in
economics, sociology, and psychology (e.g., Simmons and Sutter 2009; Ripberger et al.
2015; Trainor et al. 2015; LeClerc and Joslyn 2015; Jauernic and van den Broeke 2017;
Lim et al. 2019)

Our findings of the optimal predefined warning thresholds are similar to Roulston and
Smith (2003). Roulston and Smith (2003) developed the simple model to optimize





predefined warning thresholds considering the damage, cost, and imperfect compliance
with forecasting (i.e., the cry wolf effect). They also revealed that it is necessary to choose
high warning thresholds if intolerance of false alarms of the society is high. However,
there are substantial differences between our study and the previous cost-loss analysis
such as Roulston and Smith (2003). First, Roulston and Smith (2003) developed the static
model in which the cry wolf effect is treated exogeneously while our model is the dynamic
model in which the cry wolf effect is endogeneously simulated. Therefore, our model can
consider the temporal change in the design and accuracy of FEWS, the flood protection
level, and social systems, which may be the significant advantage to analyze the actual
socio-hydrological phenomena. Second, by fully utilizing the previous achievements of
Girons Lopez et al. (2017), we can also consider social collective memory of past
disasters, which is not considered by Roulston and Smith (2003). This feature of our
model can reveal that the social collective memory also contributes to the optimal
predefined warning thresholds.

The major limitation of this study is that our modeling of social collective trust is simple
and is not fully supported by empirical data. Although intuition and theory suggest that
many false alarms reduce the preparedness actions responding to warnings, the existence


of the cry wolf effect in the weather-related disasters is still debatable (see a
comprehensive review of Lim et al. (2019)). Simmons and Sutter (2009) indicated that
the recent false alarms negatively impacted the preparedness actions, so that we modeled
the change in social collective trust by the recent forecast outcome. However, Ripberger
et al. (2015) could not find the statistically significant short-term effect of false alarms
although they found the statistically significant cry wolf effect using the long-term data.
It should be noted that most of previous studies related to the cry wolf effect focused on
tornado disasters and the systematic econometric analyses have not been implemented for
flood disasters. The effect of social collective memory on catastrophic disasters in the
actual society is also debatable (e.g., Fanta et al. 2019). As Mostert (2018) suggested, it
is crucially important to perform case study analyses, obtain empirical data, and integrate
those data into the dynamic model to deepen our understanding of the hypothesis of the
models (e.g., Roobavannan et al. 2017; Ciullo et al. 2017; Barendrecht et al. 2019;
Sawada and Hanazaki 2020).

In socio-hydrology, researchers have mainly focused on the functions of land use change
and water-related infrastructures such as dams, levees, and dikes in the complex social
systems. Although the interactions between social systems and weather forecasting such





as the cry wolf effect are interesting, the function of FEWS and weather-related disaster
forecasting has not been intensively investigated in socio-hydrology. We call for the new
research regime, socio-meteorology, as extension of socio-hydrology. In socio-
meteorology, researchers may focus on how social systems interact with water-related
disaster forecasting, how the efficiency of weather forecasting is affected by the other
hydrological factors such as land use and flood protection infrastructures, and how
weather forecasting affects the design of land use and flood protection infrastructures.




## Acknowledgements

We used the source code of Girons Lopez et al. (2017) which can be downloaded at
https://github.com/GironsLopez/prep-fews. This study does not contain any data. This
study was supported by the JST FOREST program (grant no. JPMJFR205Q).

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





**Table 1**. Summary of the outcomes of the flood early warning system. Loss by each outcome is also shown
(see also Section 2).

|  | $Q < \delta$ | $Q \geq \delta$ |
|---|---|---|
| $P < \pi$ | True negative: 0 | False negative: $D_Q$ |
| $P \geq \pi$ | False positive: $C$ | True positive: $C + D_r$ |







**Table 2.** Fixed model parameters

|  | description | equation | values |
|---|---|---|---|
| $\kappa_c$ | shape of the bivariate gamma distribution to generate river discharge timeseries | (1) | 2.5 |
| $\theta_c$ | scale of the bivariate gamma distribution to generate river discharge timeseries | (1) | 0.08 |
| $\mu_m$ | mean of prediction error | (2) | 0 |
| $\beta$ | parameter of the damage function | (3) | 0.2 |
| $\alpha_0$ | minimum residual damage fraction | (4) | 0.2 |
| $\lambda$ | social collective memory decay rate | (8) | 0.028 |
| $\chi$ | psychological shock magnitude | (8) | 1.0 |








**Table 3.** Model parameters in the experiment 1.

|  | description | equation | values |
|---|---|---|---|
| $\sigma_m$ | standard deviation of prediction error | (2) | 0.075 |
| $\mu_v$ | mean of prediction precision | (2) | 0.15 |
| $\sigma_v$ | standard deviation of prediction precision | (2) | 0.075 |
| $\delta$ | Damage threshold | (3,5) | 0.35 |
| $\eta$ | cost parameter | (6) | 0.02 |
| $\gamma$ | Parameter controlling weights of social collective memory and trust | (7) | 1 (GL model) 0.5 (SKK model) |
| $\tau_{TP}$ | Increment of trust for true positive | (9) | 0.1 |
| $\tau_{FN}$ | Increment of trust for false negative | (9) | 0.1 |
| $\tau_{FP}$ | Increment of trust for false positive | (9) | 0.1 |








**Table 4.** Model parameters in the experiment 2

| | description | equation | values | | | | | |
|---|---|---|---|---|---|---|---|---|
| | | | exp2.1 | exp2.2 | exp2.3 | exp2.4 | exp2.5 | exp2.6 |
| $\sigma_m$ | standard deviation of prediction error | (2) | 0.05 | 0.075 | 0.05 | 0.05 | 0.075 | 0.05 |
| $\mu_v$ | mean of prediction precision | (2) | 0.05 | 0.15 | 0.05 | 0.05 | 0.15 | 0.05 |
| $\sigma_v$ | standard deviation of prediction precision | (2) | 0.025 | 0.075 | 0.025 | 0.05 | 0.075 | 0.025 |
| $\delta$ | Damage threshold | (3,5) | 0.35 | 0.35 | 0.35 | 0.35 | 0.35 | 0.35 |
| $\eta$ | cost parameter | (6) | 0 | 0 | 0.1 | 0 | 0 | 0.1 |
| $\gamma$ | Parameter controlling weights of social collective memory and trust | (7) | 1 (GL model) | 1 (GL model) | 1 (GL model) | 0.5 (SKK model) | 0.5 (SKK model) | 0.5 (SKK model) |
| $\tau_{TP}$ | Increment of trust for true positive | (9) | 0.1 | 0.1 | 0.1 | 0.1 | 0.1 | 0.1 |
| $\tau_{FN}$ | Increment of trust for false negative | (9) | 0.1 | 0.1 | 0.1 | 0.1 | 0.1 | 0.1 |
| $\tau_{FP}$ | Increment of trust for false positive | (9) | 0.1 | 0.1 | 0.1 | 0.1 | 0.1 | 0.1 |







**Table 5.** Model parameters in the experiment 3

|  | description | equation | values | | | |
|---|---|---|---|---|---|---|
|  |  |  | exp3.1 | exp3.2 | exp3.3 | exp3.4 |
| $\sigma_m$ | standard deviation of prediction error | (2) | 0.05 | 0.05 | 0.05 | 0.05 |
| $\mu_v$ | mean of prediction precision | (2) | 0.05 | 0.05 | 0.05 | 0.05 |
| $\sigma_v$ | standard deviation of prediction precision | (2) | 0.025 | 0.025 | 0.025 | 0.025 |
| $\delta$ | Damage threshold | (3,5) | 0.20 | 0.20 | 0.45 | 0.45 |
| $\eta$ | cost parameter | (6) | 0.02 | 0.02 | 0.02 | 0.02 |
| $\gamma$ | Parameter controlling weights of social collective memory and trust | (7) | 1 (GL model) | 0.5 (SKK model) | 1 (GL model) | 0.5 (SKK model) |
| $\tau_{TP}$ | Increment of trust for true positive | (9) | 0.1 | 0.1 | 0.1 | 0.1 |
| $\tau_{FN}$ | Increment of trust for false negative | (9) | 0.1 | 0.1 | 0.1 | 0.1 |
| $\tau_{FP}$ | Increment of trust for false positive | (9) | 0.1 | 0.1 | 0.1 | 0.1 |







**Table 6.** Model parameters in the experiment 4.

| | description | equation | values |
|---|---|---|---|
| $\sigma_m$ | standard deviation of prediction error | (2) | 0.05 (accurate forecast) <br> 0.075 (inaccurate forecast) |
| $\mu_v$ | mean of prediction precision | (2) | 0.05 (accurate forecast) <br> 0.15 (inaccurate forecast) |
| $\sigma_v$ | standard deviation of prediction precision | (2) | 0.025 (accurate forecast) <br> 0.075 (inaccurate forecast) |
| $\delta$ | Damage threshold | (3,5) | 0.20 (green society) <br> 0.45 (technological society) |
| $\eta$ | cost parameter | (6) | 0.02 |
| $\gamma$ | Parameter controlling weights of social collective memory and trust | (7) | 1 (GL model) |
| $[\tau_{TP}, \tau_{FN}, \tau_{FP}]$ | Increment of trust for true positive, false negative, and false positive | (9) | [0.1, 0.1, 0.1] (blue lines in Figures 4a-4h) <br> [0.1, 0.1, 0.8] (orange lines in Figures 4a-4h) <br> [0.1, 0.8, 0.1] (green lines in Figures 4a-4h) |








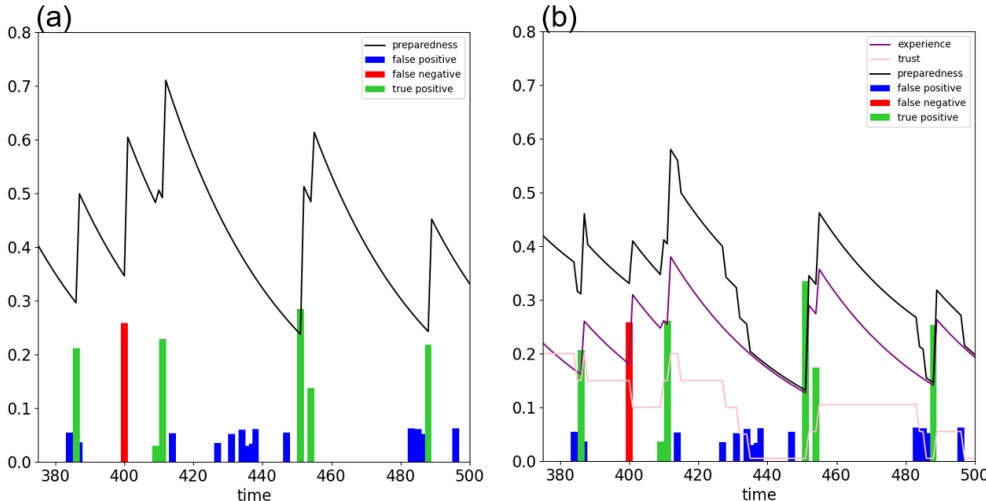


**Figure 1.** Timeseries of (a) the GL model and (b) the SKK model of the experiment 1 (see section 3 and Table

2 for model parameters). Black, purple, and pink lines are social preparedness, half of social collective memory,

and half of social collective trust in FEWS, respectively. Since social preparedness is identical to social

collective memory and social collective trust is not considered in the GL model, there are no purple and pink

lines in (a). Blue, red, and green bars show total loss by the outcomes of false positive, false negative, and true

positive, respectively.



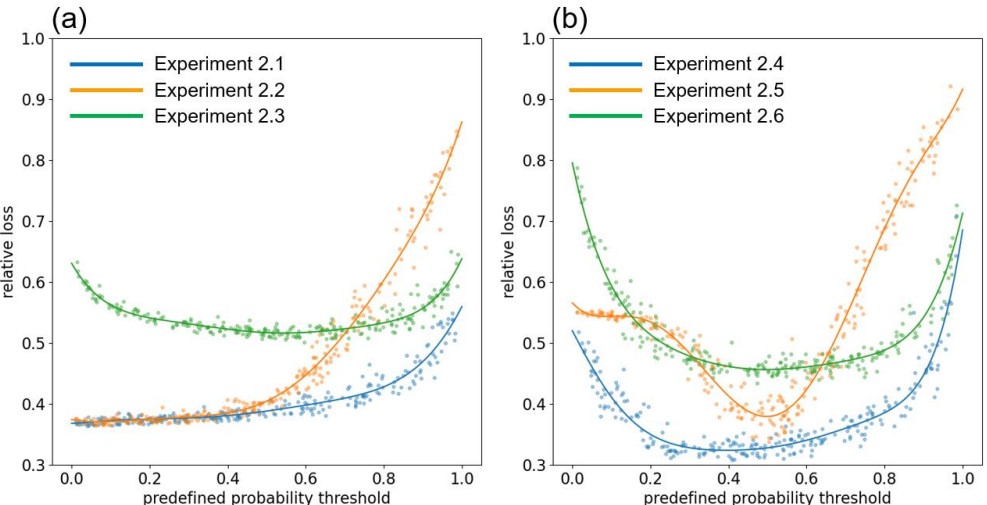

**Figure 2.** The relationship between relative loss and predefined warning thresholds in (a) the GL model and

(b) the SKK model in the experiment 2. In (a), blue, orange, and green lines show the results of the

experiments 2.1, 2.2, 2.3, respectively. In (b), blue, orange, and green lines show the results of the

experiments 2.4, 2.5, 2.6, respectively. Each dot shows the result of the individual Monte-Carlo simulation

and we smoothed them by Gaussian process regression. See also Table 4 for detailed parameter settings.



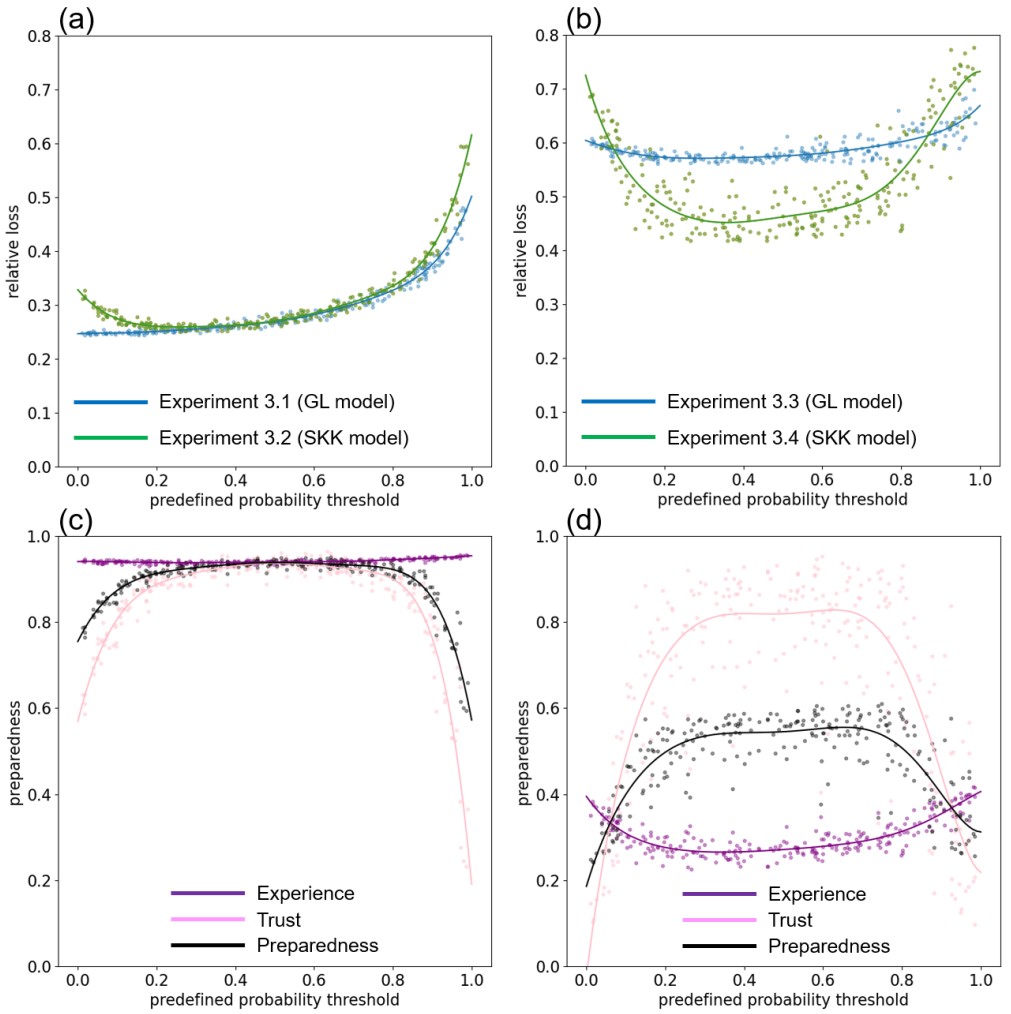

**Figure 3**. (a-b) The relationship between relative loss and predefined warning thresholds in (a) the green

society and (b) the technological society. In (a), blue and green lines show the results of the experiments 3.1

and 3.2, respectively. In (b), blue and green lines show the results of the experiments 3.3 and 3.4,

respectively. (c-d) The relationship between time-averaged social preparedness and predefined warning

thresholds in (c) the green society and (d) the technological society. Black, purple, and pink lines show time-



averaged social preparedness, social collective memory, and social collective trust in FEWS. Each dot shows
the result of the individual Monte-Carlo simulation and we smoothed them by Gaussian process regression.




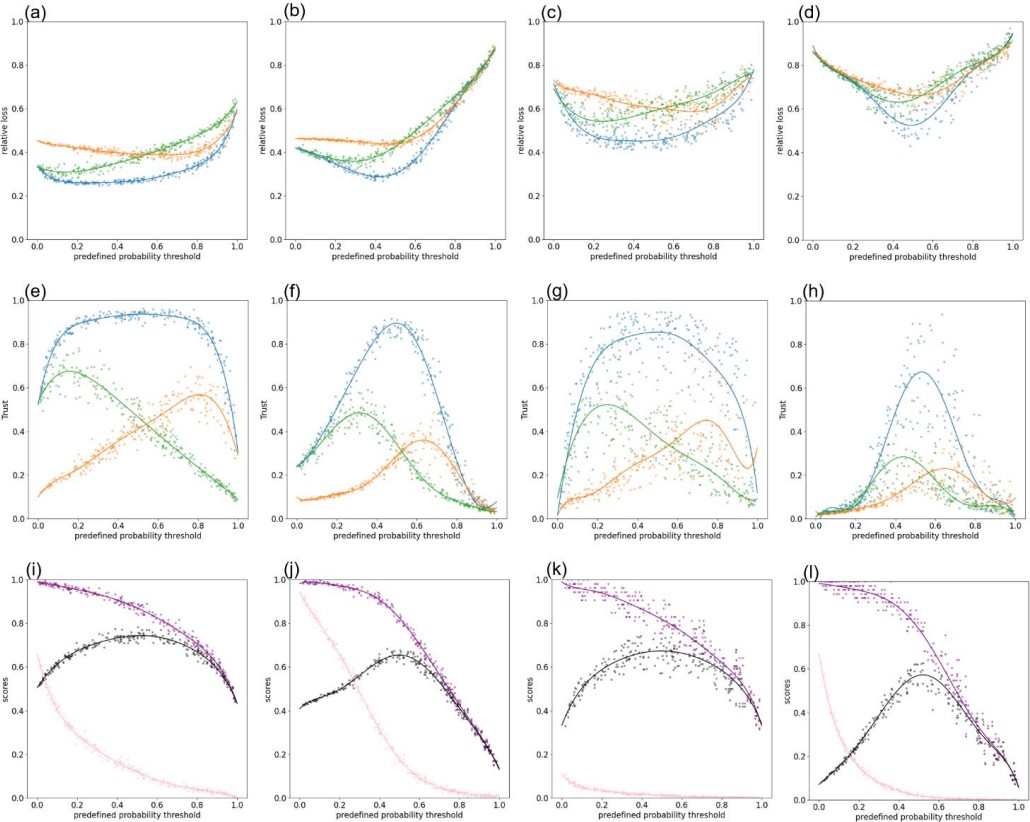


**Figure 4.** Results of the experiment 4. (a-d) The relationship between relative loss and predefined warning

thresholds in (a) the green society with accurate forecasts, (b) the green society with inaccurate forecasts, (c)

the technological society with accurate forecasts, (d) the technological society with inaccurate forecasts.

Increments of trust for true positive, false negative, and false positive are set to 0.1, 0.1, and 0.1 (blue lines),

0.1, 0.1, and 0.8 (orange lines), and 0.1, 0.8, and 0.1 (green lines). See Table 6 for detailed model

parameters' settings. (e-f) Same as (a-d) but for time-averaged social collective trust in FEWS. (i-l) Same as

(a-d) but for threat score (black lines), hit rate (purple lines), and false alarm ratio (pink lines). Each dot





shows the result of individual Monte-Carlo simulation and we smoothed them by Gaussian process

regression.

623