# Peer review of "Socio-meteorology: flood prediction, social preparedness, and cry wolf effects"

_Hydrology and Earth System Sciences, 2021_

## Author Comment (AC1)

Response letter of hess-2021-497

Dear Anonymous Referee #1,

Please find the responses to the comments.

Comments made by the reviewer were highly insightful. They allowed us to greatly improve the quality of the manuscript. We described the response to the comments.

Each comment made by the reviewers is written in *italic* font. We numbered each comment as (n.m) in which n is the reviewer number and m is the comment number. In the revised manuscript, changes are highlighted in yellow.

We trust that the revisions and responses are sufficient for our manuscript to be published in *Hydrology and Earth System Sciences*

Sincerely
Yohei Sawada, Rin Kanai, Hitomu Kotani

**Responses to the comments of Referee #1**

*This paper presents the improvement of an existing socio-hydrological model on the interactions between flood forecasting and flood loss by including social collective trust. The manuscript uses the model to investigate the cry wolf effect (where individuals may be less likely to implement protective measures if they have experienced false alarms). I believe including trust and investigating its role is a relevant contribution to the socio-hydrological literature. The manuscript shows an interesting analysis of the (potential) role of social collective trust and its implications for early warning systems.*

*(1.1) However, I believe a major limitation of the work is the lack of comparison between model results and data or empirical evidence. I appreciate that there may not be enough data available to actually compare the model results to data, but given this limitation I believe the model equations and parameter choices should be much better substantiated with evidence from the literature. In addition, one could, in a descriptive way, compare the results with findings in the literature related to the cry wolf effect rather than only compare the results to the results of another model. In the current state, the manuscript does not provide enough evidence for the model assumptions and their relevance. This means that it is impossible to draw any useful conclusions from the results of the analysis, since it is unclear how well the model represents reality.*

→ First, our model and findings are qualitatively consistent to empirical evidence found in previous works. This point has been clarified in the revised version of the paper. See also our responses to the comment of Referee #2 (2.11).

> "Our stylized model and findings are consistent to the previous works. In our model, the subjective perceptions of warning system's accuracy controls social collective trust in a weather agency and preparedness actions, which is consistent to Ripberger et al. (2015). Our simulation results reveal that more actual false alarms hamper preparedness actions and induce more damages, which is consistent to the findings of Simmons and Sutter (2009) and Trainor et al. (2015). The behavior of the optimal warning threshold is similar to Roulston and Smith (2003)."

Second, the comparison between our SKK model and the GL model in Figure 2 actually shows that our SKK model is more consistent to the published literature at least qualitatively. Figure 2 indicates that in the original GL model, it is necessary to reduce the number of false alarms to minimize the total loss only when the cost of mitigation and protection actions responding to issued warning (C in equation 6) is large. On the other hand, the reduction of false alarms is always necessary to minimize the loss in the SKK model. We found that previous works revealed that this cost is negligibly small compared with the total loss of flood disasters. Based on the fact that the mitigation cost is negligible, and forecasters take care of reducing false alarms, our extension of the GL model improves the

consistency of the simulation to reality. This point was indeed unclear in the original version of the paper. We have firstly mentioned that C is negligibly small according to the previous literature in the revised version of the paper.

> "Note that this cost has been found to be negligibly small compared with avoidable damage. For instance, Schroter et al. (2008) showed that the cost $C$ is approximately 2 % of avoidable damage. In previous works, this cost was often neglected (e.g., Pappenberger et al. 2015; Hallegatte 2012). Although Gironz Lopez et al (2017) assumed that there are non-negligible costs of mitigation and protection actions, we will discuss how differently their model and our newly proposed model works with no mitigation costs (i.e. $\eta = 0$) as well as the original settings of Gironz Lopez et al (2017)."

> "Pappenberger, F., Cloke, H. L., Parker, D. J., Wetterhall, F., Richardson, D. S., Thielen, J.: The monetary benefit of early flood warnings in Europe. Environmental Science & Policy, 51, 278-291, https://doi.org/10.1016/j.envsci.2015.04.016, 2015"

> "Schroter, K., et al: Effectiveness and efficiency of early warning systems for flash-floods (EWASE). First CRUE ERA-Net Common Call – Effectiveness and efficiency of non-structural flood risk management measures, 132pp. available from www.crue-eranet.net, 2008"

> "Hallegatte, S.: A cost effective solution to reduce disaster losses in developing countries Hydro-meteorological services, early warning, and evaculation, The World Bank Policy Research Working Paper, 6058, available from https://openknowledge.worldbank.org/bitstream/handle/10986/9359/WPS6058.pdf?s, 2012"

Then, we have clarified that the SKK model can simulate the behavior of forecasters and the relationship between warning thresholds and total losses more realistically than the GL model in the results section of the revised paper.

> "Note that the costs of mitigation and protection actions with $\eta = 0.1$ in the experiment 2.3 is comparable to the flood damages. As discussed above, this high cost of mitigation and protection actions was not supported by previous works although Girons Lopez et al. (2017) used this parameter."

> "Considering that the previous works indicated that the cast of mitigation and protection actions is negligibly small (i.e. it is realistic to assume $\eta = 0$), the SKK model reproduces the relationship between warning thresholds and total losses more realistically than the GL model."

We also briefly mentioned this point in the discussion section of the revised paper.

> "While the GL model realistically simulate the behavior of the optimal warning threshold only when unrealistically high costs of mitigation and protection actions are assumed, our stylized model needs no costs of mitigation and protection actions to realistically simulate the behavior

of the optimal warning threshold. Our stylized model is more consistent to the previous works in which the costs of mitigation and protection actions responding warnings were found to be negligibly small (e.g., Schroter et al. 2008; Hallegatte 2012; Pappenberger et al. 2015)."

*Some other remarks:*

*(1.2) The authors use socio-meteorology in their title and in the final paragraph of the discussion and conclusion they call for a new field called socio-meteorology. However, it is not clear to me why this work is so different that it does not fit within the field of socio-hydrology (the authors are only using discharge and forecasts of discharge in their model, to me this is hydrology, not meteorology). I would suggest to choose a different title, and stick to using socio-hydrology, as the authors do throughout the entire manuscript (the socio-meteorology is in fact only mentioned as an afterthought in the final paragraph of the manuscript).*

→ We propose a new title "Impact of cry wolf effects on social preparedness and efficiency of flood early warning systems". We believe this version of the title directly show what we investigated in this paper.

The reviewer mentioned that using discharge and forecasts of discharge is hydrology and is not meteorology. We disagree with this comment. To provide "forecasts" of discharge, weather forecasting is absolutely necessary, which we believe is in the field of meteorology (or it is often called as hydro-meteorology). We essentially intended to investigate the function of "forecasts" of discharge in the dynamics of social preparedness, so that we are now going to the interdisciplinary field which includes hydrology, meteorology, and social sciences.

*(1.3) Introduction, lines 43 to 80: after reading the introduction for the first time I had the impression that there was actually no evidence for the cry wolf effect and for a relationship between the false alarm ratio and the implementation of measures. This made me wonder what the relevance of the presented model and manuscript is. However, after re-reading I see that I misinterpreted and there are studies that do find evidence in support of the cry wolf effect, but also some that do not. I would suggest the authors rewrite this part of the introduction to better present the argument for why their study is important.*

→ We believe that many previous works found and quantified the cry wolf effects in meteorological disasters. In addition, many forecasters may believe the cry wolf effects when they design early warning systems. Therefore, it is crucially important to consider them. These points were indeed

unclear in the original version of the paper, and we have clarified them in the revised version of the paper by performing many modifications in this paragraph. Note that we carefully reviewed previous works which contradicts with each other in the balanced way, which we believe is still appropriate and essentially unchanged in the revised version of the paper.

"Many previous studies have found and quantified the cry wolf effects in meteorological disasters."

"Roulston and Smith (2003) found that the warning threshold of the actual weather warning systems can be justified only if the cry wolf effect is considered. This finding implies that many forecasters believe the existence of the cry wolf effects and the design of early warning systems was substantially affected by how the cry wolf effects are considered."

"It should be noted that while these previous works supported the cry wolf effect as an important factor to be considered for the design of warning systems, some studies discussed the myth of cry wolf effects implying that they do not exist."

"Although Trainor et al. (2015) supported the existence of the cry wolf effects, they also found that there is a wide variation in public definition of false alarms and actual false alarm ratio does not predict perception of false alarm ratio."

" Although the existence of the cry wolf effect is still debatable due mainly to the lack of field data and the ambiguity of the quantification of public perception of false alarms, the current evidence suggests the importance of understanding the effect of false alarms on behavioral responses to warning to design efficient flood early warning systems."

*(1.4) In the model description in line 148 (and after) the authors talk about preparedness actions (and mitigation and protection actions), please elaborate and explain what these actions are. There are many preparedness actions that do not depend on a flood warning to be implemented, what about those actions? These kind of measures may actually be implemented when experience of damage is high and trust in flood warning is low (which is the opposite of the cry wolf effect).*

→ Here we modelled preparedness actions which were done by responding issued warnings such as evacuation and safekeeping of assets. We fully agree with the reviewer's comment that many preparedness actions are unrelated to early flood warning. These preparedness actions are not included in our model to focus only on the impact of social preparedness on the efficiency of early flood warning. This point was indeed unclear in the original version of the paper. We have clarified this point in the revised version of the paper.

"If a flood event is successfully forecasted and a warning is issued (i.e. $P \geq \pi$), this damage is mitigated by preparedness actions such as evacuation and safekeeping of assets. Note that preparedness actions which are not triggered by FEWS were not considered in this stylized model

*(1.5) Equation 6 models the cost of mitigation and protection actions, why is this relevant? Please discuss why you calculate this. Later, in section 3.1, I see that the total loss is calculated as D + C. I suggest to move this to section 2, since it is quite important and now it is a bit hidden away, which means the importance of C is unclear. Also how are the costs of protection actions determined? What is this based on? Also, why is the loss calculated as D+C, please explain this.*

→ Although Table 1, which appears in section 2, shows how to calculate the total loss, this point should have been emphasized when C is introduced. We have clarified this point in the revised version of the paper.

"==Whenever a warning is issued, $C$ is included in the total loss.==”

As we discussed in our responses to the comment (1.1), we think that this cost is essentially unnecessary and should be neglected. We need to describe it just because the original GL model used it and it is important in their model. Please see our responses to the comment (1.1). We have not further modified the paper responding to this comment.

*(1.6) In lines 177 to 179, the authors state that it is reasonable to assume that trust in FEWS increases (decreases) when prediction succeeds (fails). Please elaborate, this is the main contribution of the manuscript and this claim should be substantiated more. (The authors reference Wachinger et al. (2013), but Wachiger et al. (2013) actually hypothesise that the cry wolf effect may be an explanation for the risk perception paradox and do not provide the evidence to support this hypothesis.)*

→ In the original version of the paper, the previous sentence explain why we decided to choose this simple model.

"Previous studies pointed out that the recent forecast accuracy and false alarm ratio affected the performance of preparedness actions (Simmons and Sutter 2009; Trainor et al. 2015; Ripberger et al. 2015; Jauernic and van den Broeke 2017).”

We fully agree that Wachiger et al. pointed out that the behavior of individual risk perception is more complicated than expected. We did not originally want to cite this paper here. In the revised version of the paper, we have simply deleted this reference. In addition, we clarified that our stylized model is consistent to the published literature related to cry wolf effects in the discussion section of the revised paper. See also our responses to (1.1).

"==Our stylized model and findings are consistent to the previous works. In our model, the==

subjective perceptions of warning system's accuracy controls social collective trust in a weather agency and preparedness actions, which is consistent to Ripberger et al. (2015). Our simulation results reveal that more actual false alarms hamper preparedness actions and induce more damages, which is consistent to the findings of Simmons and Sutter (2009) and Trainor et al. (2015). The behavior of the optimal warning threshold is similar to Roulston and Smith (2003)."

*(1.7) In lines 200 to 202 the authors state: "In our proposed model, high social collective trust in FEWS can maintain the high level of social preparedness even if a community completely loses past flood experiences (equation (7))." To me it seems unlikely that preparedness stays high solely based on trust while people have forgotten about floods. Is there any evidence from the literature that supports this assumption?*

→ We agree with this reviewer's comment. Here we intended to demonstrate how the new model works using some extreme cases, which we believe was misleading. Theoretically, people take preparedness actions when E goes to 0 but T is high (see equation 7). However, this situation rarely happens in our model. Social collective trust increases when disasters are predicted. In this case, social collective memory also increases because disasters happen. Therefore, E and T are somewhat correlated in many cases, and the combination of zero E and high T may not happen. What we would like to say is that the negative effect of small E can be partially mitigated by high T. We have deleted the description of unrealistic extreme cases and relaxed this sentence in the revised version of the paper.

"If social preparedness is determined only by social collective memory as Girons Lopez et al (2017) proposed, small social collective memory directly results in insufficient social preparedness actions. In our proposed model, high social collective trust in FEWS can induce social preparedness actions even if a community loses past flood experiences to some extent (equation (7))."

*(1.8) For all variables and parameters: what are the units?*

→ Units were indeed unclear in the original version of the paper. We have clarified them in the tables of the revised paper.

*(1.9) For all equations and values the authors choose: please provide more evidence from the literature as to why this is a good representation of reality. This is especially important given the lack of data for comparison with model results, as mentioned in my main point.*

→ All equations but equations 7 and 9 come from Girons Lopez et al. (2017). One can find more evidence which supports each equation in Girons Lopez et al. (2017) and references therein. We would like to avoid repeating the detailed explanation in this paper for brevity. Although this point was already mentioned in the original version of the paper, we have slightly modified the sentence to emphasize this point.

> "For brevity, the detailed explanation of equations shared with Girons Lopez et al. (2017) is omitted in this paper. See Gironz Lopez et al. (2017) and references therein for the complete description including the empirical evidence which supports each equation."

The explanation of the other equations has been strengthened by responding to the other comment. Currently no empirical evidence can support the equation 7 with $\gamma = 0.5$ very well although this chose of the equation and parameter is useful to analyze the essential behavior of our proposed model. This specific issue was pointed out by Referee #2. We attached the comment and our responses below.
* * *
*(2.6) Line 239: Why did you set gamma = 0.5? Why exactly 0.5? What does it mean?*

→ We believe that the meaning of $\gamma = 0.5$ was clearly explained in this sentence.

> Lines: In this paper, this original model is hereafter called the GL model. On the other hand, when we set $\gamma = 0.5$ in equation (7), our model considers both social collective memory and social collective trust in FEWS with same weights to calculate social preparedness.

Although this choice of $\gamma$ is somewhat arbitrary because there is no knowledge about the importance of social collective trust to induce preparedness actions compared to social collective memory. Assuming the same weights give us the most straightforward interpretation of the contributions of both factors to social preparedness and the total loss by floods since we do not have to consider the asymmetric contributions of two factors in equation (7). This point was indeed unclear in the original version of the paper, and we have clarified this point in the revised version of the paper.

> Lines: There is no existing knowledge about the relative importance of social collective memory and social collective trust. Assuming the same weights give us the most straightforward interpretation of the contributions of social collective trust and memory to social preparedness and the total loss by floods since we do not need to consider asymmetric contributions of the two factors in equation (7). Therefore, $\gamma = 0.5$ is appropriate to analyze the essential behavior of our proposed model. This new model with $\gamma = 0.5$ is hereafter called the SKK model.
* * *
*(1.10) Table 2 and lines 207- 208: why are those parameters fixed and why do they have those values? Are they based on anything?*

→ The fixed parameters are not important in our analyses. We simply choose the values which are consistent to the previous work. This point was indeed unclear in the original version of the paper, and we have clarified it in the revised version of the paper.

> "These parameters are not focused on our analysis, and we chose their values from the previous works."

The values of $\kappa_c$, $\theta_c$, $\alpha_0$, and $\chi$ are same as Girons Lopez et al. (2017). We have clarified this point in the revised version of the paper.

> "The values of $\kappa_c$, $\theta_c$, $\alpha_0$, and $\chi$ are same as Girons Lopez et al. (2017)."

$\mu_m = 0$ means the forecast is unbiased, which was mentioned in the original version of the paper.

> "While Girons Lopez et al. (2017) changes $\mu_m$ in their simulation, we set $\mu_m = 0$ assuming the forecast is unbiased."

We have mentioned once again here in the revised version of the paper.

> "We set $\mu_m = 0$ assuming the forecast is unbiased (see also equation 2 and its description)."

Although the value of $\beta$ was chosen somewhat arbitrary, it was in range specified by the original model (Girons Lopez et al. 2017). Also, the results of Girons Lopez et al. (2017) indicated that this parameter is not very sensitive to relative loss. This point was indeed unclear in the original version of the paper. We have clarified this point in the revised version of the paper.

> "Our specified $\beta$ is within the range proposed by Girons Lopez et al. (2017). In addition, the results of Girons Lopez et al. (2017) indicated that this parameter is not sensitive to relative loss."

We set $\lambda$ assuming social collective memory has 25-year half-life, which is within the range of previous works which quantified this half-life by empirical data. This point was indeed unclear in the original version of the paper, and we have clarified it in the revised version of the paper.

> "We set $\lambda$ assuming that social collective memory has 25-year half-life which is within the range of previously quantified values (e.g., Fanta et al. 2019; Barendrecht et al. 2019)."

*(1.11) For the parameters that are varied, why those values?*
→ In this study, we did not intend to mimic the real-world phenomena. Our purpose of the numerical experiments is to understand the behavior of our newly proposed stylized model. The effect of changes in parameters on the optimal warning threshold is more important than their values themselves. However, we realized that the strategy of changing parameters to understand the model's behavior has not been clear enough in the original version of the paper. We have addressed this issue in the revised

version of the paper. In the experiment 2, the prediction skill was controlled by $\sigma_m$, $\mu_v$, and $\sigma_v$. We prepared two sets of the parameters for relatively accurate and inaccurate prediction systems. We have explained this point in the revised paper.

"The prediction skill is controlled by $\sigma_m$, $\mu_v$, and $\sigma_v$. The greater values of these parameter provide inaccurate prediction. We prepared two sets of the parameter for relatively accurate and inaccurate prediction system (see Table 4)"

Please see our responses to the comment (1.1) for the discussion of the cost parameter $\eta$ in the experiment 2. We used $\eta = 0.1$ which was used in the original GL model as well as $\eta = 0$ which we believe is more consistent to the published literature. We have clarified this point in the revised paper.

"Following the settings of Girons Lopez et al. (2017), we set $\eta = 0.1$. In addition, we also performed the numerical simulation with $\eta = 0$ (i.e. negligible costs of mitigation and protection actions) which is more consistent to the published literature than the original settings (see section 2)."

In the experiment 3, we mimic the hypothetical "green" and "technological" societies by changing $\delta$. From the original value in Girons Lopez et al. (2017), we decreased and increased $\delta$ to mimic the green and technological societies, respectively. This point has been clarified in the revised paper.

"From the original value of the damage threshold proposed by Girons Lopez et al. (2017) (i.e. $\delta = 0.35$), we decreased and increased $\delta$ to simulate the green and technological societies, respectively (see Table 5)."

In the experiment 4, we focused on the responses of our proposed model to the parameters in the dynamics of social collective trust ($\tau_{TP}$, $\tau_{FN}$, and, $\tau_{FP}$ in equation (9)). We added a sentence to clarify this point in the revised version of the paper.

"We analyze how the optimal warning threshold is changed by changing $\tau_{FN}$ and, $\tau_{FP}$ (see Table 6)."

*(1.12) Figure 1: what does half of social collective trust and social collective memory mean? Why half?*

→ This is because total social preparedness is calculated as 0.5E+0.5T in our settings. If purple and pink lines are summed up, one can obtain black line. This point was mentioned in the original version of the paper.

"In this paper, this original model is hereafter called the GL model. On the other hand, when we

set $\gamma = 0.5$ in equation (7), our model considers both social collective memory and social collective trust in FEWS with same weights to calculate social preparedness."

However, we realized that this point needs to be clarified in the caption of Figure 1. We have clarified this point in the revised version of the paper.

> " Note that the sum of half of social collective memory and half of social collective trust in FEWS is social preparedness in (b)."

*(1.13) In line 289 it is stated that figure 2 shows predefined warning threshold, but the figure axis title is predefined probability threshold. Same for figures 3 and 4.*

→ We have fixed this point in the revised version of the paper.

---

## Author Comment (AC2)

Response letter of hess-2021-497

Dear Anonymous Referee #2,

Please find the responses to the comments.

Comments made by the reviewer were highly insightful. They allowed us to greatly improve the quality of the manuscript. We described the response to the comments.

Each comment made by the reviewers is written in *italic* font. We numbered each comment as (n.m) in which n is the reviewer number and m is the comment number. In the revised manuscript, changes are highlighted in yellow.

We trust that the revisions and responses are sufficient for our manuscript to be published in *Hydrology and Earth System Sciences*

Sincerely
Yohei Sawada, Rin Kanai, Hitomu Kotani

**Responses to the comments of Referee #2**

*This paper presents an advanced modelling approach to studying the relationship between society and natural extremes. The authors developed an older model by Girons Lopez et al. (2017) by adding the "cry wolf effect" phenomenon. I find this a stimulating extension.*

*As mathematical modelling is not my field of expertise, I cannot assess the modelling section (which is the core of the paper). Therefore, my review is more focused on the context and overall results.*

*The paper is well-written and readable. However, some aspects should be explained better (see below). I also pose several questions for the discussion.*

*Questions and comments:*

*(2.1) Title: Please consider changing the title to more reflect the paper's topic. E.g. "Possibilities of mathematical modelling in socio-meteorology: flood prediction etc." or "Mathematical modelling of the cry wolf effect".*

→ We propose a new title "Impact of cry wolf effects on social preparedness and efficiency of flood early warning systems". We believe this version of the title directly show what we investigated in this paper.

*(2.2) Line 68: You mention that some studies claim that the cry wolf effect does not exist. How do your results affect this debate? Why do you think some authors have found the cry wolf effect problematic? Please discuss these questions.*

→ First, field data which are useful to analyze the cry wolf effects are still lacking. It is necessary to collect the wide range of data which have an enough variance of false alarm ratios. Second, although many researchers tried to verify that citizens tend to take preparedness actions such as evacuation when they believe that a false alarm ratio of their warning system is low, the public perception of false alarm ratios is complicated and often contradicts to the actual false alarm ratio. This ambiguity of the public definition of false alarms makes the analysis complicated. These points were indeed unclear in the original version of the paper. We have clarified this point in the revised version of the paper.

"Although the existence of the cry wolf effect is still debatable due mainly to the lack of field data and the ambiguity of the quantification of public perception of false alarms, the current evidence suggests the importance to understand the effect of false alarms on behavioral responses to warning to design efficient flood early warning systems."

*(2.3) Lines 78-80: I agree with this sentence. However, it would be nice to give a practical recommendation – what does your research imply? How should we consider the social aspects?*

→ Although our practical implications are clear, we hesitated to explicitly mention them based on our preliminary modelling work in the original version of the paper. We have clarified them in the discussion and conclusion section of the revised version of the paper.

"Although our model is simple and stylized, we can provide practically useful implication to improve the design of FEWS. First, considering the dynamics of social collective trust in FEWS is more important in the technological society with infrequent flood events than in the green society with frequent flood events. It implies that weather agencies need more efforts to be trusted by general citizens to induce their preparedness actions when a community is protected by flood protection infrastructures such as levees and dams more heavily. Second, as the natural scientific skill to predict flood is improved, the efficiency of FEWS gets more sensitive to the behavior of social collective trust, so that forecasters need to determine their forecasting threshold by considering the social aspects. Considering the recent advances of the skill to predict extreme hydrometeorological events, this finding implies that it is becoming more important for forecasters to take social dynamics responding to weather forecasts into consideration."

*(2.4) Line 139: You mention that the "trust in flood warnings" is based on the accuracy of warnings. Are there any other possibilities how to increase public trust? E.g. by social activities, education etc.? I understand that the mathematical model must be simple, but please discuss this.*

→ We fully agree with this reviewer's comment. Trust in FEWS may be significantly impacted by social activities and education, which we neglected in our stylized model. This point was indeed unclear in the original version of the paper. We have clarified this point in the revised version of the paper:

"Note that although there are many other factors which affect social collective trust in FEWS such as social activities and education, we did not consider them in our stylized model."

*(2.5) Lines 163-164:    Please define the terms "social collective memory" and "social collective trust".*

→ The definition of them in this work was indeed unclear in the original version of the paper. Social collective memory is the shared knowledge and information about past flood disasters occurred in a community. We have clarified it in the revised version of the paper.

"Social collective memory is shared knowledge and information about past flood disasters occurred in a community."

The definition of social collective trust is analogous to social collective memory. It is shared knowledge and perception of the reliability of the information issued from authorities. We have clarified it in the revised version of the paper.

"Social collective trust is defined as shared knowledge and perception of the reliability of the information issued from authorities. "

*(2.6) Line 239: Why did you set gamma = 0.5? Why exactly 0.5? What does it mean?*

→ We believe that the meaning of $\gamma = 0.5$ was clearly explained in this sentence.

"In this paper, this original model is hereafter called the GL model. On the other hand, when we set $\gamma = 0.5$ in equation (7), our model considers both social collective memory and social collective trust in FEWS with same weights to calculate social preparedness."

Although this choice of $\gamma$ is somewhat arbitrary because there is no knowledge about the importance of social collective trust to induce preparedness actions compared to social collective memory. Assuming the same weights give us the most straightforward interpretation of the contributions of both factors to social preparedness and the total loss by floods since we do not have to consider the asymmetric contributions of two factors in equation (7). This point was indeed unclear in the original version of the paper, and we have clarified this point in the revised version of the paper.

"There is no existing knowledge about the relative importance of social collective memory and social collective trust. Assuming the same weights give us the most straightforward interpretation of the contributions of social collective trust and memory to social preparedness and the total loss by floods since we do not need to consider asymmetric contributions of the two factors in equation (7). Therefore, $\gamma = 0.5$ is appropriate to analyze the essential behavior of our proposed model. This new model with $\gamma = 0.5$ is hereafter called the SKK model."

*(2.7) Line 257: Please explain the terms "green society" and "technological society".*

→ We believe that the definitions are explained right after this sentence. In the revised version of the paper, we have added more details.

"In green society, risk is dealt with mainly by non-structural measures. In this society, the flood

protection level is so low that many flood events occur, which increases social collective memory of flood events. In technological society, the flood protection level is so high that risk can be dealt with by structural measures as well as non-structural measures. Since flood events occur less frequently in the technological society, the high level of social collective memory cannot be maintained."

*(2.8) Tables 2, 3, 4, 5, 6: Please explain how you got the values of the parameters. Are these values based on empirical knowledge or literature review? Or are they just selected arbitrarily? What do these values mean?*

→ Referee #1 also pointed out this point. How to specify the fixed parameters shown in Table 2 and how to change the other parameters in Tables 3-6 can be found in the revised version of the paper. We attached the comments of Referee #1 and our responses below, which we believe address this comment (2.8).
* * *
*(1.10) Table 2 and lines 207- 208: why are those parameters fixed and why do they have those values? Are they based on anything?*

→ The fixed parameters are not important in our analyses. We simply choose the values which are consistent to the previous work. This point was indeed unclear in the original version of the paper, and we have clarified it in the revised version of the paper.

"These parameters are not focused on our analysis, and we chose their values from the previous works."

The values of $\kappa_c$, $\theta_c$, $\alpha_0$, and $\chi$ are same as Girons Lopez et al. (2017). We have clarified this point in the revised version of the paper.

"The values of $\kappa_c$, $\theta_c$, $\alpha_0$, and $\chi$ are same as Girons Lopez et al. (2017)."

$\mu_m = 0$ means the forecast is unbiased, which was mentioned in the original version of the paper.

"While Girons Lopez et al. (2017) changes $\mu_m$ in their simulation, we set $\mu_m = 0$ assuming the forecast is unbiased."

We have mentioned once again here in the revised version of the paper.

"We set $\mu_m = 0$ assuming the forecast is unbiased (see also equation 2 and its description)."

Although the value of $\beta$ was chosen somewhat arbitrary, it was in range specified by the original model (Girons Lopez et al. 2017). Also, the results of Girons Lopez et al. (2017) indicated that this parameter is not very sensitive to relative loss. This point was indeed unclear in the original version

of the paper. We have clarified this point in the revised version of the paper.

"Our specified $\beta$ is within the range proposed by Girons Lopez et al. (2017). In addition, the results of Girons Lopez et al. (2017) indicated that this parameter is not sensitive to relative loss."

We set $\lambda$ assuming social collective memory has 25-year half-life, which is within the range of previous works which quantified this half-life by empirical data. This point was indeed unclear in the original version of the paper, and we have clarified it in the revised version of the paper.

"We set $\lambda$ assuming that social collective memory has 25-year half-life which is within the range of previously quantified values (e.g., Fanta et al. 2019; Barendrecht et al. 2019)."

*(1.11) For the parameters that are varied, why those values?*

→ In this study, we did not intend to mimic the real-world phenomena. Our purpose of the numerical experiments is to understand the behavior of our newly proposed stylized model. The effect of changes in parameters on the optimal warning threshold is more important than their values themselves. However, we realized that the strategy of changing parameters to understand the model's behavior has not been clear enough in the original version of the paper. We have addressed this issue in the revised version of the paper. In the experiment 2, the prediction skill was controlled by $\sigma_m$, $\mu_v$, and $\sigma_v$. We prepared two sets of the parameters for relatively accurate and inaccurate prediction systems. We have explained this point in the revised paper.

"The prediction skill is controlled by $\sigma_m$, $\mu_v$, and $\sigma_v$. The greater values of these parameter provide inaccurate prediction. We prepared two sets of the parameter for relatively accurate and inaccurate prediction system (See Table 4)"

Please see our responses to the comment (1.1) for the discussion of the cost parameter $\eta$ in the experiment 2. We used $\eta = 0.1$ which the original GL model also used as well as $\eta = 0$ which we believe is more consistent to the published literature. We have clarified this point in the revised paper.

"Following the settings of Girons Lopez et al. (2017), we set $\eta = 0.1$. In addition, we also performed the numerical simulation with $\eta = 0$ (i.e. negligible costs of mitigation and protection actions) which is more consistent to the published literature than the original settings (see section 2)."

In the experiment 3, we mimic the hypothetical "green" and "technological" societies by changing $\delta$. From the original value in Girons Lopez et al. (2017), we decreased and increased $\delta$ to mimic the green and technological societies, respectively. This point has been clarified in the revised paper.

"From the original value of the damage threshold proposed by Girons Lopez et al. (2017) (i.e.

$\delta = 0.35$), we decreased and increased $\delta$ to simulate the green and technological societies, respectively (see Table 5)."

In the experiment 4, we focused on the responses of our proposed model to the parameters in the dynamics of social collective trust ($\tau_{TP}$, $\tau_{FN}$, and, $\tau_{FP}$ in equation (9)). We added a sentence to clarify this point in the revised version of the paper.

"We analyze how the optimal warning threshold is changed by changing $\tau_{FN}$, and, $\tau_{FP}$ (see Table 6)."
* * *
*(2.9) Figure 1: Why do you show precisely the time range you show? Is it a random selection? What does mean the height of the colour bars? Is it the flood intensity or damage level? Would you please add a description of the y-axis?*

→ It is nearly a random selection, but we chose the time range which clearly shows the difference between the SKK and GL models. The purpose of the experiment 1 and Figure 1 is to demonstrate how differently the SKK and GL models work. This purpose of the experiment 1 was mentioned in the original version of the paper.

"In the experiment 1, the timeseries of state variables of the two models are compared to demonstrate how differently the SKK and GL models work."

In the revised version of the paper, we have emphasized this purpose when we explained Figure 1.

"The purpose of Figure 1 is to demonstrate how differently the SKK and GL models work by showing the timeseries."

The height of the color bars shows total loss by the outcomes (see Table 2), which was mentioned in the original version of the paper. We refer to Table 2 to make it easier to understand this point in the revised version of the paper.

"Blue, red, and green bars show total loss by the outcomes of false positive, false negative, and true positive, respectively (see Table 2)."

We have added a description of the y-axis in the revised version of the paper.

*(2.10) Discussion and conclusion: According to your findings, it is possible to give a practical recommendation to FEWS strategy? I.e. do you suggest issuing fewer warnings (to reduce the cry wolf*

*effect but risk the damages of flooding) or more warnings (to be safer but risk the cry wolf effect)? Please discuss.*

→ In the revised version of the paper, we have explicitly mentioned that our findings are practically useful, and the practical implications have been explained. See our response to the comment (2.3).

We believe that our stylized model has a potential to assess the appropriate warning threshold if it can be accurately localized. This point was indeed unclear in the original version of the paper, and we have clarified this point in the revised version of the paper by briefly describing:

> "Similar to Roulston and Smith (2003), our stylized model has a potential to help forecasters determine the optimal warning threshold if it can be appropriately calibrated by empirical data."

We avoid directly suggesting issuing fewer warnings or more warnings in this paper. It strongly depends on the local conditions. It is too risky to say it based on our current findings.

*(2.11) Discussion and conclusion: Please also discuss your findings in the context of papers on the cry wolf effect you mentioned in the Introduction section.*

→ Our stylized model and findings are consistent to the previous works cited in the Introduction section. This point was indeed unclear in the original version of the paper, and we have clarified it in the revised version of the paper. In the original paper, we explained that our stylized model cannot fully support the existing empirical data in one paragraph. We added the paragraph to comprehensively explain the similarity and differences from previous works.

> "Our stylized model and findings are consistent to the previous works. In our model, the subjective perceptions of warning system's accuracy controls social collective trust in a weather agency and preparedness actions, which is consistent to Ripberger et al. (2015). Our simulation results reveal that more actual false alarms hamper preparedness actions and induce more damages, which is consistent to the findings of Simmons and Sutter (2009) and Trainor et al. (2015). The behavior of the optimal warning threshold is similar to Roulston and Smith (2003)."

*(2.12) Final remarks: Your paper is based on the modelling approach only. Would you please suggest how it would be possible to validate your findings on real data?*

→ This comment was mostly identical to the comment of Referee #1. Although it is not currently possible to directly compare our results with empirical data, our proposed model is consistent to the published literature at least qualitatively. The response to the comment (1.1) is attached below.
* * *
*1.1) However, I believe a major limitation of the work is the lack of comparison between model results and data or empirical evidence. I appreciate that there may not be enough data available to actually compare the model results to data, but given this limitation I believe the model equations and parameter choices should be much better substantiated with evidence from the literature. In addition, one could, in a descriptive way, compare the results with findings in the literature related to the cry wolf effect rather than only compare the results to the results of another model. In the current state, the manuscript does not provide enough evidence for the model assumptions and their relevance. This means that it is impossible to draw any useful conclusions from the results of the analysis, since it is unclear how well the model represents reality.*

→ First, our model and findings are qualitatively consistent to empirical evidence found in previous works. This point has been clarified in the revised version of the paper. See also our responses to the comment of Referee #2 (2.11).

> "Our stylized model and findings are consistent to the previous works. In our model, the subjective perceptions of warning system's accuracy controls social collective trust in a weather agency and preparedness actions, which is consistent to Ripberger et al. (2015). Our simulation results reveal that more actual false alarms hamper preparedness actions and induce more damages, which is consistent to the findings of Simmons and Sutter (2009) and Trainor et al. (2015). The behavior of the optimal warning threshold is similar to Roulston and Smith (2003)."

Second, the comparison between our SKK model and the GL model in Figure 2 actually shows that our SKK model is more consistent to the published literature at least qualitatively. Figure 2 indicates that in the original GL model, it is necessary to reduce the number of false alarms to minimize the total loss only when the cost of mitigation and protection actions responding to issued warning (C in equation 6) is large. On the other hand, the reduction of false alarms is always necessary to minimize the loss in the SKK model. We found that previous works revealed that this cost is negligibly small compared with the total loss of flood disasters. Based on the fact that the mitigation cost is negligible, and forecasters take care of reducing false alarms, our extension of the GL model improves the consistency of the simulation to reality. This point was indeed unclear in the original version of the paper. We have firstly mentioned that C is negligibly small according to the previous literature in the revised version of the paper.

> "Note that this cost has been found to be negligibly small compared with avoidable damage. For instance, Schroter et al. (2008) showed that the cost $C$ is approximately 2 % of avoidable damage. In previous works, this cost was often neglected (e.g., Pappenberger et al. 2015; Hallegatte 2012). Although Gironz Lopez et al (2017) assumed that there are non-negligible costs of mitigation and protection actions, we will discuss how differently their model and our newly

proposed model works with no mitigation costs (i.e. $\eta = 0$) as well as the original settings of Gironz Lopez et al (2017)."

"Pappenberger, F., Cloke, H. L., Parker, D. J., Wetterhall, F., Richardson, D. S., Thielen, J.: The monetary benefit of early flood warnings in Europe. Environmental Science & Policy, 51, 278-291, https://doi.org/10.1016/j.envsci.2015.04.016, 2015"

"Schroter, K., et al: Effectiveness and efficiency of early warning systems for flash-floods (EWASE). First CRUE ERA-Net Common Call – Effectiveness and efficiency of non-structural flood risk management measures, 132pp. available from www.crue-eranet.net, 2008"

"Hallegatte, S.: A cost effective solution to reduce disaster losses in developing countries Hydro-meteorological services, early warning, and evaculation, The World Bank Policy Research Working Paper, 6058, available from https://openknowledge.worldbank.org/bitstream/handle/10986/9359/WPS6058.pdf?s, 2012"

Then, we have clarified that the SKK model can simulate the behavior of forecasters and the relationship between warning thresholds and total losses more realistically than the GL model in the results section of the revised paper.

"Note that the costs of mitigation and protection actions with $\eta = 0.1$ in the experiment 2.3 is comparable to the flood damages. As discussed above, this high cost of mitigation and protection actions was not supported by previous works although Girons Lopez et al. (2017) used this parameter."

"Considering that the previous works indicated that the cast of mitigation and protection actions is negligibly small (i.e. it is realistic to assume $\eta = 0$), the SKK model reproduces the relationship between warning thresholds and total losses more realistically than the GL model."

We also briefly mentioned this point in the discussion section of the revised paper.

"While the GL model realistically simulate the behavior of the optimal warning threshold only when unrealistically high costs of mitigation and protection actions are assumed, our stylized model needs no costs of mitigation and protection actions to realistically simulate the behavior of the optimal warning threshold. Our stylized model is more consistent to the previous works in which the costs of mitigation and protection actions responding warnings were found to be negligibly small (e.g., Schroter et al. 2008; Hallegatte 2012; Pappenberger et al. 2015)."
* * *
One of the major obstacles for the validation of the model is the lack of data. Although the lack of data was mentioned in the original version of the paper, we did not explicitly say that the lack of data makes it difficult to validate the model. We have clarified this point in the revised version of the paper.

"It should be noted that most of previous studies related to the cry wolf effect focused on tornado disasters and the systematic econometric analyses have not been implemented for flood disasters, which makes it difficult to validate our proposed model."

---

## Author Response (AR2)

Response letter of hess-2021-497

Dear Editor and Anonymous Referee #1,

Please find the responses to the comments.

Comments made by the reviewer were highly insightful. They allowed us to greatly improve the quality of the manuscript. We described the response to the comments.

Each comment made by the reviewer is written in *italic* font. We numbered each comment as (n.m) in which n is the reviewer number and m is the comment number. In the revised manuscript, changes are highlighted in yellow.

We trust that the revisions and responses are sufficient for our manuscript to be published in *Hydrology and Earth System Sciences*

Sincerely
Yohei Sawada, Rin Kanai, Hitomu Kotani

**Responses to the comments of Referee #1**

*I am not entirely satisfied with the revision of the paper in reply to both reviewers' comments. Specifically:*

*(1.1) Comment 1.6: the authors' response here is that the their assumption about trust is explained with the sentence: "Previous studies pointed out that the recent forecast accuracy and false alarm ratio affected the performance of preparedness actions (Simmons and Sutter 2009; Trainor et al. 2015; Ripberger et al. 2015; Jauernic and van den Broeke 2017)." This talks about the performance of preparedness actions, not about an increase or decrease in trust, or even about the implementation of measures. It does not explain why it is reasonable to assume that trust in FEWS increases (decreases) when prediction succeeds (fails). This needs to be better substantiated with evidence from the literature.*
→ LeClerc and Joslyn directly investigated the relationship between trust ratings and false alarms by the controlled experiment. They found that trust ratings are increased by the decreased false alarm levels. Our model of social collective trust is based on this finding. This point was indeed unclear in the original version of the paper. We have clarified this issue in the revised version of the paper.

> Lines 198-201: In the controlled experiment of LeClerc and Joslyn (2015), medium-range trust ratings are increased by decreased false alarm levels. Their experiments revealed that trust ratings are based on the pattern of forecasts and observations over the previous month.

*(1.2) Comment 1.7: in response to my point that it is not realistic that preparedness increases only because of trust while there is no memory of an event, the authors' decided to remove this discussion from the text, rather than update the model so that it is more realistic. Even if this does not happen often, this behaviour is not realistic and therefore implies a flaw in the model structure. This should be addressed or at the very least discussed as a limitation.*
→ We examined this point again, and currently we believe that $P_r(t) > 0$ with $E(t) = 0$ is not so unrealistic. In an individual level, even if people have never experienced damages by themselves, they may be able to take preparedness actions based on information from their trusted authorities. Forecasters strongly expect this behavior. If $P_r(t) = 0$ with $E(t) = 0$ in an individual level, investment in disaster prediction systems cannot be justified in most of highly protected urbanized areas such as Tokyo in which most of citizens have never experienced water levels above damage thresholds by themselves. Since $E(t)$ depends only on damages, $E(t) = 0$ does not necessarily mean that they have no memory about flood events nor they fully forget the existence about flood. $E(t)$ should be interpreted as collective *personal* experiences of flood damages and should not be interpreted as a simple memory and knowledge about flood. Many disasters prevention measures such

as education, evacuation drills, and weather forecasting are designed to help people avoid the risk of flooding even if they have no personal experiences of flood disasters. To evaluate the effectiveness of these measures, $P_r(t) = 0$ with $E(t) = 0$ is not an appropriate behavior of the model although we admit that the effectiveness of forecasting highly depends on $E(t)$ as Girons Lopez et al. (2017) discussed.

Therefore, we still believe that the current additive form of the equation (7) is appropriate even at the limit of $E(t) \to 0$ with $T(t) \gg 0$ although it rarely happens as we discussed in the previous round. In the revised version of the paper, we have justified the additive form of the equation (7) using the discussion written above:

> Lines 231-243: The additive form of the equation (7) implies that preparedness actions are taken even if either social collective memory $E(t)$ or social collective trust $T(t)$ goes to zero. Note that $E(t) \approx 0$ does not mean that a community does not know the existence of a flood event while it means most of citizens have never experienced water levels above damage thresholds by themselves. Many disasters prevention measures such as education, evaluation drills, and FEWS are designed to let people take preparedness actions even if they have no personal experiences of flood disasters. Forecasters expect that people take preparedness actions based on information from their trusted authorities even if they have never experienced damages by themselves. To evaluate the effectiveness of these measures, $P_r(t) = 0$ with $E(t) = 0$ is not an appropriate behavior of the model although the effectiveness of FEWS highly depends on $E(t)$ as Girons Lopez et al. (2017) found. Therefore, we chose the additive form of the equation (7) rather than the other simple alternatives such as multiplicative forms.

*(1.3) Comment 2.2: I believe the authors should more thoroughly discuss what their model results add to the discussion of whether the cry wolf effect exists or not. In my opinion, simply stating "the current evidence suggests the importance to understand the effect of false alarms on behavioral responses to warning in order to design efficient flood early warning systems." is not the same as discussing your results in the context of the current debate in the scientific literature.*

→ Our modeling study does not give any evidences to conclude the debate on the existence of cry wolf effects and the purpose of this study is not to reveal the existence of cry wolf effects. We successfully justified the current behavior of forecasters, which balance the number of false alarms with that of missed events, more realistically than the existing dynamic models. This point has been discussed in the original version of the paper:

> Lines 522-528: While the GL model realistically simulates the behavior of the optimal warning threshold only if unrealistically high costs of mitigation and protection actions are assumed, our

stylized model needs no costs of mitigation and protection actions to realistically simulate the behavior of the optimal warning threshold. Our stylized model is more consistent to the previous works in which the costs of mitigation and protection actions responding warnings were found to be negligibly small (e.g., Schroter et al. 2008; Hallegatte 2012; Pappenberger et al. 2015).

Note that this result implies that forecasters believe the existence of cry wolf effects, which is one of the important implications in our modeling work, but it does not necessarily mean that cry wolf effects exist. This point has been emphasized in the revised version of the paper.

> Lines 528-531: Our results justify the optimal warning thresholds which balance false alarms with missed events and imply that forecasters believe the existence of cry wolf effects, although it does not necessarily mean that cry wolf effects exist.

It is necessary to perform and accumulate more sophisticated field surveys and econometric analyses to contribute to the debate on the existence of cry wolf effects. Although our modeling work cannot directly contribute to it, we can obtain some useful implications for the design of future field surveys. First, our results show the sensitivity of relative loss to predefined probability threshold around the optimal value is small in many cases. In many field surveys such as Simmons and Sutter (2009) and Trainor et al. (2015), pairs of false alarm ratio and damage in many regions of one country are collected and compared to show the increase of false alarm ratio increases damage. Assuming that nationwide criteria of issuing warnings are almost optimal, our study implies that the observable signal of cry wolf effects in this approach is weak. It may be the reason why several field surveys contradict with each other. Our modeling results imply that it is difficult to quantify cry wolf effects using time-mean performance of warnings and damages at least for the flood disasters. We recommend analyzing the temporal change in behaviors responding to recent forecast outcomes, although this strategy seems to be costly and time-consuming.

Second, Figure 3 of the paper implies that it is better to choose technological societies as a research field because it is more difficult to distinguish the contributions of experience and trust in flood-prone areas. These points were indeed unclear in the original version of the paper. We have included them in the revised version of the paper.

> Lines 554-571: As discussed above, systematic econometric analyses and field surveys on cry wolf effects have not been implemented for flood disasters, so that it is important to design such kinds of analyses. Our modelling work provides useful implications for the design of future field analyses. First, our results show that the sensitivity of relative loss to predefined probability threshold is small around its optimal value in many cases. In many field surveys such as Simmons and Sutter (2009) and Trainor et al. (2015), pairs of false alarm ratio and damage in many regions

of one country are collected and compared to show the increase of false alarm ratio increases damage. Assuming that nationwide criteria of issuing warnings are near-optimal, our study implies that the detectable signal of cry wolf effects in this approach is weak. Our modeling work implies that it is difficult to quantify cry wolf effects using time-mean performance of warnings and damages. It may be the reason why several field surveys contradict with each other and the negative effect of false alarm ratio cannot be found in some surveys (Lim et al. 2019). We recommend analyzing the temporal change in behaviors responding to recent forecast outcomes, although this strategy is costly and time-consuming. Second, our experiment 3 implies that it is better to choose technological societies as a research field because it is more difficult to distinguish the contributions of experience and trust in less protected areas.

*(1.4) Comment 2.6: in reply to the reviewers' question about the choice of value for γ the authors reply there is no evidence from the literature on which to base this value. The justification for choosing 0.5 (i.e. not having to consider asymmetric contributions of E and T) is not good enough. The addition of trust in the model is the main contribution, if there is a parameter that determines the influence of trust on your model outcome and there is no evidence that suggests a certain value, one should at least investigate the effect of the parameter's value on the end results of the study. I would suggest to add a sensitivity analysis of the study's results to this parameter.*

→ We have added how $\gamma$ affects the relationship between predefined probability threshold and relative loss in the revised version of the paper.

Lines 292-293: The behavior of the models with the different $\gamma$ is also discussed in the supplement material.

Lines 398-404: Figure S2 shows how $\gamma$ in the equation (7) affects the relationship between relative loss and predefined probability threshold. When the contribution of social collective trust to social preparedness increases (i.e., $\gamma$ gets smaller), the "implicit cost" of false alarms induced by relatively small predefined probability thresholds increases. Figure S2 also shows that moderate changes of $\gamma$ from the default setting of the SKK model (i.e. 0.5) do not qualitatively change the relationship between relative loss and predefined probability threshold.

[Figure]

**Figure S2**. Sensitivity of $\gamma$ in equation (7) to the relationship between relative loss and predefined probability thresholds. No cost of the mitigation and protection action is assumed. In (a), the high prediction accuracy that is same as experiments 2.1 and 2.4 is assumed. In (b) the lower prediction accuracy that is same as experiments 2.2 and 2.5 is assumed. Blue, orange, green, red, and purple dots and lines are results with $\gamma = 0, 0.25, 0.5, 0.75, and\ 1$, respectively. Each dot shows the result of the individual Monte-Carlo simulation and we smoothed them by Gaussian process regression. See also Table 4 for detailed parameter settings.

*(1.5) Comment 2.9: while the chosen time range may clearly show the difference between the two models, it would also be good to include a figure that shows the behaviour of the model over the entire time range.*

→ We have shown it in a supplement material.

 Lines 347-349: While Figure 1 shows the subset of the entire timeseries to clearly demonstrate the differences between two models, the entire timeseries can be found in Figure S1 of the supplement material.

[Figure]

**Figure S1.** Same as Figure 1 but for the entire time range.

*(1.6) Comment 1.1, 2.11, 2.12: The authors do now discuss their results in relation to the literature that they discuss in the introduction, but only in relation to the literature that supports the existence of the cry wolf effect. Why do the other papers find that this effect does not exist? And how do the results of this study relate to that? How do they support the evidence that the effect does exist?*

→ For example, the survey design of Trainor et al. (2015) and Lim et al. (2019) is similar, but the conclusions are completely different. Especially, the contribution of actual false alarm ratio is debating in the literature. As discussed in our response to the comment (1.3), we found that the sensitivity of relative loss to predefined warning threshold is small around the optimal value of the threshold. It implies that it is difficult to obtain the signal of cry wolf effect in the real world, which may be able to explain the contradiction of previous works. This point has been included in the revised version of the paper. See also our response to the comment (1.3).

Lines 557-566: First, our results show that the sensitivity of relative loss to predefined probability threshold is small around its optimal value in many cases. In many field surveys such as Simmons and Sutter (2009) and Trainor et al. (2015), pairs of false alarm ratio and damage in many regions of one country are collected and compared to show the increase of false alarm ratio increases damage. Assuming that nationwide criteria of issuing warnings are near-optimal, our study

implies that the detectable signal of cry wolf effects in this approach is weak. Our modeling work implies that it is difficult to quantify cry wolf effects using time-mean performance of warnings and damages. It may be the reason why several field surveys contradict with each other and the negative effect of false alarm ratio cannot be found in some surveys (Lim et al. 2019).

*(1.7) In addition, after reading both reviewer's comments and the author's replies I tried to reproduce the results presented in the paper and was unable to do this with the information provided in the paper. Therefore, I have some further points that I believe should be addressed before the paper can be published:*

→ Thank you very much for checking the details of the study. We have made our source code publicly available. See https://gitlab.com/ysawada/sociometeorology as well as our responses to the comments below.

Lines 590-592: **Code and Data Availability**

The code to perform the numerical experiments is available in a public repository (https://gitlab.com/ysawada/sociometeorology).

*(1.8) Using a gaussian distribution for the variance of the forecast implies you get negative values, which is not possible, how is this solved? Is a truncated distribution used? If so, this should be mentioned.*

→ A truncated distribution was used. This point was indeed unclear in the original version of the paper, and we have clarified this point in the revised version of the paper:

Lines 128-129: Negative $N(\mu_v, \sigma_v^2)$ is truncated to 1.0e-6 to prevent from obtaining negative values of variance.

*(1.9) Similarly, in general (even when assuming the above mentioned variance is set to be >0), the forecast distribution yields negative forecasts for the discharge. This does not make sense, the discharge cannot be negative.*

→ We fully agree with this comment. We did not deal with this issue. It does not affect the consequences of the dynamic model since the concern is if forecasted flood level is above the (positive) damage threshold. However, this is the limitation of our model due to the oversimplification of a forecasting system and should be discussed. In the revised version of the paper, we have mentioned this issue.

Lines 133-136: Although this simplification of the forecasting system unrealistically assigns nonzero probability to negative values of discharge, it does not affect the process dynamics since the model evolution depends only on whether forecasted discharge is above the damage threshold, as we explain in the next paragraph.

*(1.10) In addition, the parameter settings used for the discharge imply a discharge of 0.2 corresponds to the 2.4 year flood and a discharge of 0.45 corresponds to a 21 year flood. This last value, 0.45, is used as the damage threshold for the technological society, but a protection level protecting against a 21 year flood is rather low for a technological society. I would expect it to protect at least against a 100 year flood (i.e. a damage threshold between 0.6 and 0.61)*

→ We believe that the reviewer's expectation of technological society, which is to fully protect at least against a 100-year flood, is too severe. In Japan, most of the largest river basins do not achieve this level. If the reviewer's criterion for technological society is applied, we guess few technological societies exist in Asian monsoon regions which have severer rainfall than European countries. To check how the models work with a larger damage threshold, we performed additional experiments with a damage threshold of 0.60 and Figure R1 shows the result. It does not change the take home messages in the experiment 2. The notably difference from the original experiments is that 1000-year averaged relative loss are strongly affected by random processes (see dots in Figure R1). This is because the number of floods in the 1000-year simulation is very small and the forecast outcomes of this several events fully determined the overall performances, which increases sampling errors. Although we decided not to include Figure R1 in the paper, we have discussed this point in the revised version of the paper.

Lines 434-437: These behaviors of the models can be found when damage threshold is further increased to 0.6, although the 1000-year averaged statistics are strongly affected by random processes due to the insufficient number of disaster events within the 1000-year computation period (not shown).

[Figure]

**Figure R1.** Same as Figure 3a or 3c, but for damage threshold of 0.6.

*(1.11) The initial values for E and T are not reported. And related to that: do they remain the same across all experiments or do you vary them? The initial values may have a significant effect on the model results and therefore the conclusions.*

→ The initial conditions of E and T are randomly selected from 0-1. This point was written in the section 3.2 of the original version of the paper:

> Lines 336-338: In experiments 2–4, we performed the 250-member Monte-Carlo simulation by randomly perturbing a predefined probability threshold, $\pi$, and the initial conditions of social collective memory and social collective trust in FEWS. We analyzed the sensitivity of the efficiency of FEWS to predefined probability thresholds.

Although we randomly choose 250 initial conditions, we used the same 250 combinations of initial conditions for all experiments. The initial conditions do affect the model results, which in part was shown in the variance of dots in our Figures 2-4, although this effect becomes small by integrating the

model for 1000-year and averaging this long-term simulation results. However, the difference of initial conditions does not affect the differences between experiments (e.g., between experiments 2.1 and 2.2) since we used the same combination of initial conditions. This point was indeed unclear in the original version of the paper, and we have clarified that the differences between experiments do not depend on any random processes in the revised version of the paper.

> Lines 338-340: We used the same random seed to generate 250-member Monte-Carlo simulation in each experiment, so that the differences between experiments do not depend on random processes.

Lastly, the initial conditions of the experiment 1 are also randomly chosen, and we have explicitly written the values of them in the revised version of the paper.

> Lines 297-298: The initial conditions of $E$ and $T$ are randomly chosen and set to 0.49 and 0.77, respectively.

*(1.12) Similarly to the initial values, the order of the floods will influence the model results. Running the model for several different floods time series and then averaging the outcome metric (e.g. the relative loss) over those runs yields more robust results.*

→ We have performed the experiments with 10 different discharge timeseries, and Figure S3 shows the results. We found that the uncertainty induced by the order of the floods is comparable to the uncertainty quantified by 250 Monte-Carlo simulations with different initial conditions and forecast outcomes. Our take home messages are qualitatively unchanged by different discharge timeseries sampled from the same gamma distribution. Since we believe that it is straightforward to compare different experiments under a single sequence of events and the accurate estimation of the outcome metric is not the main purpose of this theoretical study, we still use a single timeseries in the revised version of the paper. However, we have clarified the discussion written here using Figure S3 in the main manuscript of the revised paper.

> Lines 404-407: In addition, the qualitative behavior of our SKK model is robust to different discharge timeseries (Figure S3). Figure S3 reveals that the uncertainty induced by different discharge timeseries is comparable to that quantified by 250 Monte-Carlo simulations with different initial conditions and forecast outcomes.

[Figure]

Figure S3. The relationship between relative loss and predefined probability thresholds in (a) the experiment 2.4 and (b) the experiment 2.5. In (a), the blue line shows the original result shown in Figure 2(b) and black lines show the results with 10 different river discharge timeseries which are sampled from the gamma distribution shown in the equation (1). In (b), the orange line shows the original result shown in Figure 2(b) and black lines show the results with 10 different river discharge timeseries which are sampled from the gamma distribution shown in the equation (1). Each dot shows the result of the individual Monte-Carlo simulation and we smoothed them by Gaussian process regression. See also Table 4 for detailed parameter settings.

*(1.13) In the form as it is written in the paper, the trust T explodes in a negative or positive direction, rather than being constrained between 0 and 1. Similarly the experience E is not constrained to be below 1. Therefore the preparedness can become very large or negative, giving negative damage.*

→ We totally agree with this comment. We constrained E and T to 0-1. This point was indeed unclear in the original version of the paper, and we have clarified this point in the revised version of the paper.

Lines 190-191: When $E$ becomes larger than 1, it is truncated to 1.

Lines 205-206: When $T$ becomes larger than 1, it is truncated to 1. When $T$ becomes smaller than 0, it is truncated to 0.

*(1.14) For most experiments conducted and reported in the paper, $\pi$ is varied from 0 to 1. But for the first comparison between the two models, a fixed value is used. What is this value?*

→ It was set to 0.40. We have clarified this point in Table 3 of the revised version of the paper.